# SLAP:
# SHORTCUT LEARNING FOR ABSTRACT PLANNING

**Y. Isabel Liu**[†]   **Bowen Li**[*]   **Benjamin Eysenbach**[†]   **Tom Silver**[†]

[†]Princeton University   [*]Carnegie Mellon University
isabel.liu@princeton.edu

## ABSTRACT

Long-horizon decision-making with sparse rewards and continuous states and actions remains a fundamental challenge in AI and robotics. Task and motion planning (TAMP) is a model-based framework that addresses this challenge by planning hierarchically with abstract actions (options). These options are manually defined, limiting the agent to behaviors that we as human engineers know how to program (pick, place, move). In this work, we propose Shortcut Learning for Abstract Planning (SLAP), a method that leverages existing TAMP options to automatically discover new ones. Our key idea is to use model-free reinforcement learning (RL) to learn *shortcuts* in the abstract planning graph induced by the existing options in TAMP. Without any additional assumptions or inputs, shortcut learning leads to shorter solutions than pure planning, and higher task success rates than flat and hierarchical RL. Qualitatively, SLAP discovers dynamic physical improvisations (e.g., slap, wiggle, wipe) that differ significantly from the manually-defined ones. In experiments in four simulated robotic environments, we show that SLAP solves and generalizes to a wide range of tasks, reducing overall plan lengths by over 50% and consistently outperforming planning and RL baselines.

Project repository: https://github.com/isabelliu0/SLAP

## 1 INTRODUCTION

Long-horizon embodied tasks are fundamentally challenging for modern model-free decision-making systems (Mao et al., 2024) due to sparse rewards, complex physical interactions, and the need for generalization in continuous state and action spaces. Task and motion planning (TAMP) (Garrett et al., 2021; Dantam et al., 2016; Srivastava et al., 2014; Kaelbling & Lozano-Pérez, 2011; Toussaint, 2015; Lin et al., 2023) is a classical, model-based framework that uses state and action abstractions to meet these challenges. However, most existing TAMP systems rely on pre-defined skills (options) such as pick, place, and move, which make strong assumptions about the physical interactions between the agent and the environment (Wang et al., 2021; Mandlekar et al., 2023; Liang et al., 2024). As a result, agents are limited to behaviors that we as human engineers know how to manually program.

For example, consider the task shown in Figure 1, where a tower of obstacles must be disassembled so that a target block can be placed on a specific region. Blocks-world tasks like this one have been used to benchmark planning methods for decades (Slaney & Thiébaux, 2001; Ghasemipour et al., 2022), and it is now easy to find a plan that unstacks the obstacles one-by-one before picking and placing the target block. This solution is satisficing (Röger & Helmert, 2010), but also long and inefficient. A clever child would find a better one: pick up the target block immediately, then "slap" the obstacle tower aside before placing the target. Such a short and dynamic solution is beyond the capabilities of TAMP and other classical planners, which typically assume that the agent makes contact with objects only through its fingertips (Billard & Kragic, 2019) and that each skill influences only a small, pre-specified set of objects (cf. the STRIPS assumption (Fikes & Nilsson, 1971)).

How can an intelligent agent autonomously improvise skills that transcend traditional assumptions in robot programming and lead to better (shorter) overall plans? Previous work in hierarchical reinforcement learning (RL) has considered discovering options from *low-level* environment cues, often with entropy-based objectives (Kulkarni et al., 2016; Andrychowicz et al., 2017; Nachum et al.,

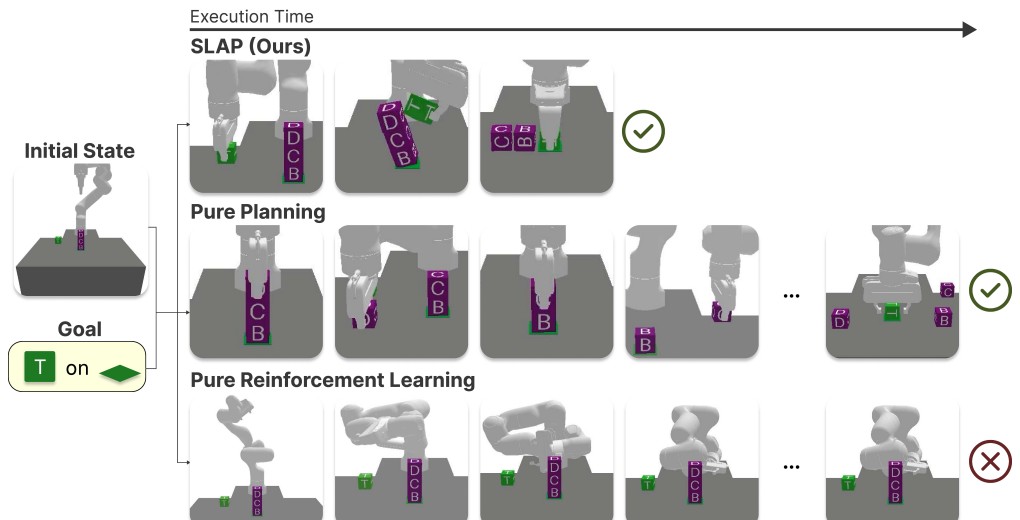

Figure 1: **Shortcut Learning for Abstract Planning (SLAP)** uses reinforcement learning (RL) to find low-level shortcuts in abstract plans. SLAP finds shorter trajectories than pure planning and achieves higher success rates than pure RL.

2018; Haarnoja et al., 2018; Eysenbach et al., 2019; Savinov et al., 2018). These *tabula rasa* methods have had limited success in the long-horizon robotic manipulation tasks that motivate our work.

Rather than learning new skills from scratch, we are interested in the practical setting where a limited set of manually defined skills is available and we wish to learn new ones. Our key insight is that the *high-level* structure of existing skills can guide learning new skills that yield better (shorter) plans. We propose Shortcut Learning for Abstract Planning (SLAP), a method that uses RL to learn new options in the abstract planning graph induced by existing skills. Specifically, SLAP identifies promising shortcut connections between abstract states and instantiates RL option-learning environments with goal-based rewards. At inference time, SLAP leverages these learned options to generate shorter plans. For example, in the blocks task, after picking up the target block with a given skill, the robot uses the learned slap shortcut to clear the target region, then place the block on the target.

From a user's perspective, SLAP is a plug-and-play module: whenever one seeks to improve the execution efficiency of an abstract planner in a given domain, SLAP can autonomously learn shortcuts without additional user input. In the extremes, if shortcuts are too difficult to learn, SLAP reduces to pure planning; if the tasks are easy, SLAP reduces to pure RL—the plan collapses into a single shortcut. In between, SLAP automatically navigates this spectrum between planning and learning.

We evaluate SLAP in four robotic environments featuring long horizons, sparse rewards, and complex physical interactions. Across all environments, SLAP consistently achieves higher success rates than flat and hierarchical RL, and shorter execution times than pure planning. In additional analyses, we find that the number of shortcuts discovered by SLAP increases throughout training time, yielding commensurate improvements in output plan lengths, and that SLAP can generalize to tasks with new and different numbers of objects than seen during training. To the best of our knowledge, SLAP is the first method that learns low-level skills for improving the execution time of an abstract planner. This represents progress toward a unified system with the improvisational flexibility of RL and the long-horizon reasoning and generalization capabilities of TAMP.

## 2 RELATED WORK

**Task and Motion Planning.** Task and Motion Planning (TAMP) combines high-level symbolic reasoning with continuous geometric motion planning to solve long-horizon, complex robotic tasks. *Task planning* decomposes unstructured, long-horizon problems into smaller symbolic subproblems (Fikes & Nilsson, 1971; Bonet & Geffner, 2001), while *motion planning* finds collision-free paths via sampling (Kavraki et al., 1996; LaValle & Kuffner Jr, 2001; Karaman & Frazzoli, 2011) or

trajectory optimization (Ratliff et al., 2009; Schulman et al., 2014). A significant body of work in robotics studies the tight coupling between task planning and motion planning (Kaelbling & Lozano-Pérez, 2011; Dantam et al., 2016; Toussaint, 2015; Srivastava et al., 2014; Garrett et al., 2020). Our abstract planner is intentionally simple—without heuristics or continuous skill optimization—to isolate shortcut learning, though more TAMP techniques can be integrated to get orthogonal benefits.

**Learning for Task and Motion Planning.** Our work is related to recent efforts that combine ideas from TAMP and machine learning. Previous works have considered learning state abstractions (Silver et al., 2023; Han et al., 2024; Shah et al., 2024; Asai & Fukunaga, 2018; Ahmetoglu et al., 2022; Li et al., 2025) and action abstractions (Silver et al., 2022; 2021b; Cheng & Xu, 2023; Agia et al., 2023; Mandlekar et al., 2023; Kokel et al., 2021; Yang et al., 2018; Lee et al., 2022; Illanes et al., 2020) to make TAMP possible. We instead assume that these abstractions are given and focus on learning to improve upon them. Other works have considered using learning to accelerate TAMP, e.g., by learning heuristics (Driess et al., 2020; Chitnis et al., 2016), object-based abstractions (Silver et al., 2021a; Zhang et al., 2024), or compiled policies (McDonald & Hadfield-Menell, 2022; Dalal et al., 2023; Katara et al., 2024). These approaches accelerate the planning process itself, rather than learning new low-level behaviors for the robot, as we do here. Other works use RL to learn recovery policies that bring the robot back to an abstract state when execution diverges from an abstract plan, e.g., when something novel in the environment occurs (Jiang et al., 2018; Li et al., 2024; Vats et al., 2023; Sarathy et al., 2020; Goel et al., 2022). We instead assume that our given TAMP skills are sufficiently robust that recovery is not necessary, and focus on improving solution efficiency.

**Hierarchical Reinforcement Learning.** Our work is related to recent efforts in hierarchical RL which assume that prior knowledge about the high-level policy is available and focus on learning low-level skills (Kokel et al., 2021; Yang et al., 2018; Lee et al., 2022; Illanes et al., 2020; Jothimurugan et al., 2021; Icarte et al., 2018). In general, hierarchical RL focuses on the problem of decomposing long-horizon, complex tasks into a hierarchy of simpler subtasks, where a high-level policy selects subgoals for low-level skills to reach. Previous works bridge planning and hierarchical RL (Allen et al., 2023), e.g., by constructing goal graphs from replay buffers and using graph search to decompose tasks into reachable waypoints (Eysenbach et al., 2019; Savinov et al., 2018). In contrast, our work aims to use RL to improvise low-level behaviors for short execution times while leveraging prior knowledge from the abstract planner for task decomposition and high-level decision-making.

## 3 PROBLEM FORMULATION

Following previous work in TAMP, (e.g., see Garrett et al. (2021) for a survey), we develop our approach in fully-observable and deterministic environments with continuous states and actions; however, see Appendix D.4 for additional results with weaker assumptions. Given a state $x \in \mathcal{X}$ and action $u \in \mathcal{U}$, the next state $x' \in \mathcal{X}$ is determined by a known transition function $f : \mathcal{X} \times \mathcal{U} \to \mathcal{X}$ (e.g., a physics simulator). We consider goal-based tasks $(x_0, g)$ where $x_0 \in \mathcal{X}$ is an initial state and $g \subseteq X$ is a goal. A solution to a task is a trajectory $\tau = (x_0, u_1, x_1, \ldots, u_T, x_T)$ where $x_T \in g$ and $x_t = f(x_{t-1}, u_t)$ for all $1 \le t \le T$. Our objective is to minimize $|\tau|$, the number of time steps in $\tau$. If actions are executed at a fixed rate, this is equivalent to minimizing execution time. We consider a distribution of tasks and assume access to a set of training tasks from the distribution. The agent is allowed training time and then evaluated on held-out tasks from the same distribution.

We further suppose that the agent has access to a partitioning of the state space that we refer to as the *abstract state space*. Let $s \in \mathcal{S}$ denote an abstract state and $\mathrm{abstract}(x) = s$ denote if $x \in s$. For simplicity, we assume that each task goal $g$ is equivalent to the union of one or more abstract states: $g = \bigcup_{s_g \in \mathcal{S}_g} s_g$. A key insight from both TAMP and hierarchical RL is that abstract states can make planning easier. A typical approach (Srivastava et al., 2014; Silver et al., 2022) is to define options (Sutton et al., 1999; Eysenbach et al., 2018) that each bring the agent from one abstract state to another. An option $a \in \mathcal{A}$ is characterized by an initial abstract state $s_{\mathrm{init}}^a \in \mathcal{S}$, a terminal abstract state $s_{\mathrm{term}}^a$, and a policy $\pi^a : \mathcal{X} \to \mathcal{U}$. When the option is initiated in $x$ such that $\mathrm{abstract}(x) = s_{\mathrm{init}}^a$, the policy $\pi^a$ is executed until the terminal abstract state $s_{\mathrm{term}}^a$ is reached. We assume that a given finite set of options $\mathcal{A}$ is sufficient for generating solutions for goals in our task distribution. However, these solutions will often be highly suboptimal with respect to execution time. We are interested in using training to learn to improve on execution time during evaluation.

## 4 SLAP: SHORTCUT LEARNING FOR ABSTRACT PLANNING

We now describe Shortcut Learning for Abstract Planning (SLAP), our proposed method for learning to improve the execution time of an abstract planner. A summary of our algorithm is presented in Figure 3 and we also illustrate SLAP via pseudocode in Algorithms 1 and 2.

### 4.1 PLANNING WITH ABSTRACT STATES

We begin by considering planning: given a task $(x_0, g)$ and options $\mathcal{A}$, how can we find solutions that minimize execution time? We propose to build and search within an *abstract planning graph* (Figure 2). The graph has two levels. In the top level, nodes represent abstract states and edges represent options. In the bottom level, nodes represent environment states and edges represent environment actions. The levels are related in that bottom-level edges correspond to top-level edge executions. To build the graph, we start at the root nodes ($x_0$ on the bottom and $\mathrm{abstract}(x_0)$ on the top) and simulate options given the known transition function. We build the graph breadth-first until we reach some nodes where the goal is satisfied. Given a built graph, we can run any shortest path algorithm (e.g., Dijkstra's) in the bottom level to find an execution-time-minimizing solution. This abstract planning graph is constructed similarly to the bilevel graphs used in previous works (Silver et al., 2022; 2023; Li et al., 2025). See Appendix A.1 for details.

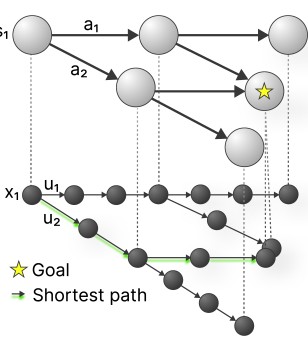

Figure 2: Abstract planning graph. Top level has abstract states $s$ and options $a$. Bottom level has states $x$ and actions $u$.

### 4.2 LEARNING SHORTCUTS WITH RL

The trajectories found by planning with the given options may be highly suboptimal, especially if the options were designed with strong simplifying assumptions about robot contact and single-object manipulation (Billard & Kragic, 2019). We propose that the agent should use training to learn *shortcuts* between abstract states to discover new low-level behaviors that may reduce execution time. A shortcut is an option $\hat{a} = \langle s_{\mathrm{init}}, \pi_\theta, s_{\mathrm{term}} \rangle$ where $s_{\mathrm{init}}$ and $s_{\mathrm{term}}$ are a pair of abstract states not already achieved by any given option, and $\pi_\theta$ is a policy with learnable parameters $\theta \in \mathbb{R}^n$. The shortcut in Figure 1 uses a learned "slap" policy to get from an abstract state where the target block is held to an abstract state where the target region is clear.

During training, we spawn multiple self-contained environments and learn shortcut policies in parallel. The environment for a shortcut from $s_{\mathrm{init}}$ to $s_{\mathrm{term}}$ is an indefinite-horizon Markov decision process (MDP) with state space $\mathcal{X}$, action space $\mathcal{U}$, transition function $f$, reward function $R(x) = -1$, and terminal states $s_{\mathrm{term}}$. To create an initial state distribution, we do not assume that we can sample directly from $s_{\mathrm{init}}$; instead, we sample from the states encountered in the abstract planning graphs for the training tasks. Given this setup, we can use any continuous-state-and-action RL algorithm to learn shortcut policies (see Appendix B.2 for ablations). We use proximal policy optimization (PPO) (Schulman et al., 2017).

---

**Algorithm 1:** SLAP Training

**Data Collection (offline)**
> **input**: $\{(x_0, g)\}, f, \mathcal{A}, N_{\mathrm{collect}}$
> **init**: $\mathcal{D} \leftarrow \{\}$                  // shortcut data
> **foreach** $(x_0, g)$ **do**
>> **for** $i = 1$ **to** $N_{collect}$ **do**
>>> $\mathcal{G} \leftarrow \mathrm{BUILDGRAPH}(x_0, g, f, \mathcal{A})$
>>> $\hat{\mathcal{D}} \leftarrow \mathrm{GETSHORTCUTDATA}(\mathcal{G})$
>>> $\hat{\mathcal{D}} \leftarrow \mathrm{PRUNE}(\hat{\mathcal{D}}, f)$           // rollouts
>>> $\mathcal{D}.\mathrm{update}(\hat{\mathcal{D}})$
>
> **return** $\mathcal{D}$

**Training (offline)**
> **input**: $f, \mathcal{D}$           // from data collection
> **init**: $\hat{\mathcal{A}} \leftarrow \{\}$           // learned shortcuts
> **foreach** $(s_{\mathrm{init}}, s_{\mathrm{term}}, \mathcal{X}_0) \in \mathcal{D}$ **do**
>> $\mathcal{M} \leftarrow \mathrm{CREATEMDP}(s_{\mathrm{init}}, s_{\mathrm{term}}, f)$
>> $\pi_\theta \leftarrow \mathrm{LEARNPOLICY}(\mathcal{M}, \mathcal{X}_0)$
>> $\hat{\mathcal{A}}.\mathrm{add}((s_{\mathrm{init}}, s_{\mathrm{term}}, \pi_\theta))$
>
> **return** $\hat{\mathcal{A}}$

---

**Algorithm 2:** SLAP Evaluation

**Evaluation (online)**
> **input**: $(x_0, g), f, \mathcal{A}, \hat{\mathcal{A}}$
> $\mathcal{G} \leftarrow \mathrm{BUILDGRAPH}(x_0, g, f, \mathcal{A} \cup \hat{\mathcal{A}})$
> $\tau \leftarrow \mathrm{DIJKSTRA}(\mathcal{G})$
> **return** $\tau$

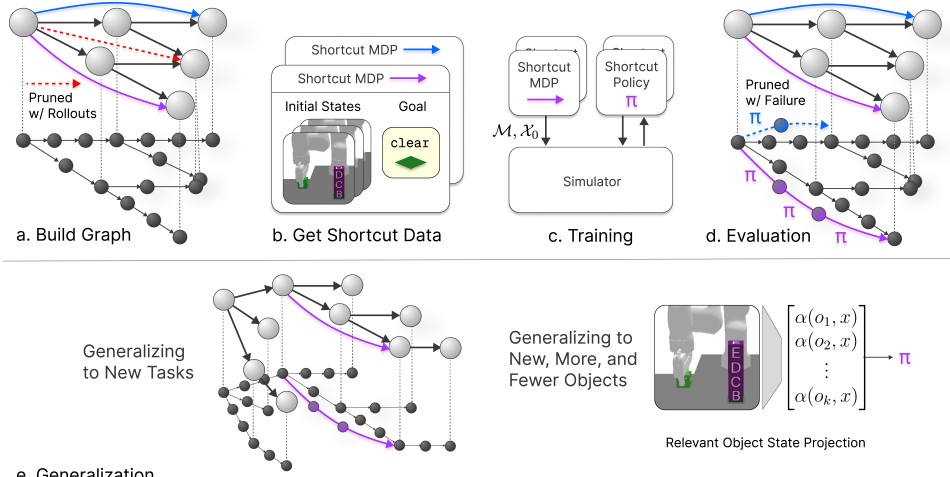

Figure 3: **SLAP Pipeline.** *(a)* We build abstract planning graphs on training tasks and generate possible shortcuts. *(b)* Each shortcut induces an MDP. *(c)* We run RL in parallel shortcut MDPs to create shortcut policies. *(d)* The learned policies are used to find shortcuts in abstract planning graphs for new evaluation tasks. *(e)* SLAP generalizes over tasks (initial states and goals) and objects.

The number of potential shortcuts is $O(|\mathcal{S}|^2)$, which can be large. We propose a simple pruning mechanism that we found to be effective in practice. For each shortcut-learning MDP, we start by executing $N_{\text{rollout}}$ random rollouts of up to length $T_{\text{rollout}}$ from the initial state. If $s_{\text{term}}$ is reached in fewer than $K_{\text{rollout}}$ rollouts, we prune the shortcut and do not run RL. The intuition behind this pruning is that RL needs some initial success to bootstrap policy learning. See Appendix B.1 for ablation studies on hyperparameter choices and additional results on the effectiveness of pruning.

## 4.3 PLANNING WITH LEARNED SHORTCUTS

Presented with a new task at evaluation, we run the same abstract planner as in Section 4.1, but with the learned shortcut policies added to the original set of options. Since shortcut policies may fail, we check if the abstract terminal state is reached within $T_{\text{eval}}$ steps and prune the edges in the case of failure (see Figure 3d). Successful shortcuts are automatically selected by the planner when they enable shorter plans. See Appendix D.2 for analysis on test-time planning and execution efficiency.

By planning with learned shortcuts, we can generalize to new tasks that have initial states and goals not seen during training. Generalization over states is achieved by the shortcut policies. This low-level generalization need not be perfect—if some shortcuts work in some tasks, we will already see benefits over pure planning. Generalization over goals is achieved by the planner, which runs search for each new goal and selects shortcuts accordingly (see Appendix D.1 for results).

Overall, our method—Shortcut Learning for Abstract Planning (SLAP)—allows us to navigate the spectrum between pure planning and pure RL. If the given options $\mathcal{A}$ are already optimal, or if shortcut learning is too hard, SLAP reduces to pure planning. If the environment is simple enough for RL, and shortcut policies can be learned directly from the initial state to the goal, SLAP reduces to pure RL. In other cases, SLAP automatically discovers a middle ground between planning and RL.

## 4.4 GENERALIZING OVER OBJECTS

TAMP methods typically assume that states are defined by *objects* and *relations* (Garrett et al., 2021). In this section, we show how SLAP can leverage the same assumption to generalize over objects, solving held-out tasks with new, more, and fewer objects than those seen during training.

Following previous work (Diuk et al., 2008; Silver et al., 2022; 2023), we suppose that each state $x \in \mathcal{X}$ is defined by a set of objects $\mathcal{O}$ and feature vectors $\alpha(o, x) \in \mathbb{R}^m$ for each object $o \in \mathcal{O}$. For example, the features for a block in Figure 1 include $x$ position and yaw orientation, among others.

We also suppose that each abstract state $s \in \mathcal{S}$ is defined by a finite set of atoms, which are discrete relations between objects, e.g., $\{\mathrm{on}(B,C), \mathrm{on}(C,D), \ldots, \mathrm{holding}(A)\}$.

During shortcut learning, we first use the relations in the abstract states to decide which atoms and objects are *relevant* for each shortcut (Silver et al., 2022). For a shortcut $\hat{a} = \langle s_{\mathrm{init}}, \pi_\theta, s_{\mathrm{term}} \rangle$, let $\mathrm{add}(\hat{a})$ be the set of atoms present in $s_{\mathrm{term}}$ but absent in $s_{\mathrm{init}}$, and let $\mathrm{del}(\hat{a})$ be those in $s_{\mathrm{init}}$ but not in $s_{\mathrm{term}}$. For example, if $\hat{a}$ corresponds to grasping object $B$, then $\mathrm{holding}(B) \in \mathrm{add}(\hat{a})$ and $\mathrm{gripperEmpty}() \in \mathrm{del}(\hat{a})$. $\mathrm{add}(\hat{a})$ and $\mathrm{del}(\hat{a})$ compose the *relevant atoms* for the shortcut, and $\mathrm{rel}(\hat{a}) \subseteq \mathcal{O}$ is the set of *relevant objects* that appear in any of the relevant atoms.

We use the relevant objects to define a state projection $\mathrm{proj}_{\hat{a}}(x) = \alpha(o_1, x) \circ \cdots \circ \alpha(o_k, x)$ where $o_i \in \mathrm{rel}(\hat{a})$ for some fixed object ordering and where $\circ$ denotes vector concatenation. When training the policy for the shortcut $\hat{a}$, we use the projected state as the observation input. As a result, adding irrelevant objects to the environment has no impact on the policy.

During evaluation, when presented with new objects, the agent considers object substitutions for each shortcut that would render the shortcut equivalent to some shortcut seen during training. Formally, given a pair of shortcuts $(\hat{a}_{\mathrm{train}}, \hat{a}_{\mathrm{eval}})$, we check if there is a type-preserving, injective object mapping $\sigma : \mathrm{rel}(\hat{a}_{\mathrm{train}}) \to \mathrm{rel}(\hat{a}_{\mathrm{eval}})$ such that

$$\{\, a_\sigma : a \in \mathrm{add}(\hat{a}_{\mathrm{train}}) \,\} \subseteq \mathrm{add}(\hat{a}_{\mathrm{eval}}), \quad \{\, a_\sigma : a \in \mathrm{del}(\hat{a}_{\mathrm{train}}) \,\} \subseteq \mathrm{del}(\hat{a}_{\mathrm{eval}}),$$

where $a_\sigma$ denotes the atom obtained by replacing each object in the atom $a$ with its image under $\sigma$. For an example of object substitution for shortcuts, see Section 5.1. If a matching object substitution is found, the respective learned shortcut policy is deployed using the substituted objects as inputs. See A.2 for details and pseudo-code.

## 5 EXPERIMENT

We next present experiments and results to address the following questions about the efficiency and effectiveness of SLAP:

**Q1.** To what extent can SLAP find shorter plans compared to pure planning?

**Q2.** How does the sample efficiency of SLAP compare to that of pure RL and hierarchical RL?

**Q3.** Does SLAP continue to improve and discover new shortcuts throughout training?

**Q4.** To what extent can SLAP generalize to new tasks and new objects?

**Q5.** Which RL design decisions are important for learning shortcuts?

Our open-source code is available online at `https://github.com/isabelliu0/SLAP`.

**System, Hardware, and Compute Footprint.** We conduct all experiments on a single H100 GPU with 4 CPU cores. Training is conducted on the same hardware as evaluation. We report the compute footprint of shortcut learning with parallelized training across subprocesses.

| Environment | # Shortcuts | Env Interactions | RL Time |
|---|---|---|---|
| Obstacle 2D | 11 | $(4.1 \pm 1.2) \times 10^6$ | $\sim$1.5 min |
| Obstacle Tower | 92 | $(4.8 \pm 0.6) \times 10^7$ | $\sim$9 h |
| Cluttered Drawer | 74 | $(3.3 \pm 0.5) \times 10^7$ | $\sim$6 h |
| Cleanup Table | 54 | $(3.2 \pm 0.6) \times 10^7$ | $\sim$8 h |

**Environments.** We evaluate our methods in four simulated robotic environments that feature long horizons, sparse rewards, and continuous states and actions. A brief overview of the environments is provided below; see Appendix B for further details and visualizations of our environments.

- **Obstacle 2D:** Inspired by the "Cover" environment (Chitnis et al., 2022; Silver et al., 2021b; 2023), a 2D planar robot with a gripper must move a target object into a designated region that is initially occupied by an obstacle. The initial options $\mathcal{A}$ implement picking and placing. Without shortcuts, the planner would pick and place the obstacle, then pick and place the target object.[1]

---

[1]Despite being in 2D, this environment is difficult for Pure RL because precise actions are required to execute grasping. We verified that weakening the threshold for grasping leads to 100% success for the Pure RL baselines.

- **Obstacle Tower:** As illustrated in Figure 1, a 7-DoF Franka Emika Panda robot arm, simulated in PyBullet (Coumans & Bai, 2016), must move a target block into a target region that is initially occupied by a tower of obstacles. The initial options implement picking and placing with BiRRT for motion planning and IKFast for inverse kinematics. Without shortcuts, the planner would pick and place each obstacle in the tower, then pick and place the target object.

- **Cluttered Drawer:** The same Franka robot simulated in PyBullet must retrieve an object from within a cluttered drawer and place it on top of a table. The initial options implement picking and placing, again with BiRRT and IKFast. Since the target object is tightly surrounded by other objects, the planner (without shortcuts) would pick and place neighboring objects until a feasible grasp exists for the target, then pick and place the target object.

- **Cleanup Table:** This PyBullet environment features realistic and irregular 3D objects from Objaverse (Deitke et al., 2023). The same Franka robot must collect three toys (duck, robot, dinosaur toys) and a wiper from the table and organize them in the storage bin on the floor beside the table. The initial options implement picking and dropping, again with BiRRT and IKFast. Without shortcuts, the planner picks up each object from the table and drops it into the bin.

**Methods Evaluated.** We now briefly describe the methods that we compare in experiments, with implementation details provided in Appendix C.

- **Shortcut Learning for Abstract Planning (SLAP)**: Our main approach.
- **Pure Planning**: The same abstract planner used by SLAP (Section 4.1), but without shortcuts.
- **Pure RL (PPO)**: Proximal policy optimization (Schulman et al., 2017) operating in the low-level joint state space of the robot on the full task with a sparse reward function that penalizes execution time (plan length). Note that SLAP shortcut learning also uses PPO.
- **Pure RL (SAC+HER)**: Given our focus on sparse-reward environments, we also compare against hindsight experience replay (HER) (Andrychowicz et al., 2017), which was designed to handle sparse rewards. We use soft actor-critic (SAC) (Haarnoja et al., 2018) as the base algorithm (which must be off-policy).
- **Hierarchical RL (PPO)**: Hierarchical RL outputs both low-level actions and skill selection probabilities. When skill activations exceed threshold 0.5, the top skill is executed until completion; otherwise, low-level actions are used. Similar to SLAP, Hierarchical RL has access to predefined skills, following modular HRL methods like MAPLE (Nasiriany et al., 2022), which dynamically select and compose behavior primitives to solve long-horizon manipulation tasks.
- **SOL**: Based on state-of-the-art hierarchical RL method Scalable Option Learning (Henaff et al., 2025), we adapted the algorithm to also have access to both predefined skills and SLAP's shortcut data. SOL jointly learns a high-level controller that selects between predefined skills and shortcuts, and low-level shortcut policies. Predefined skills are frozen throughout training, same as the high-level priors that SLAP and Hierarchical RL (PPO) leverage, while SLAP's shortcut data provide the intrinsic rewards that SOL requires.

**Experimental Details.** We begin by collecting training tasks and graphs as outlined in Algorithm 1. We sample 10 tasks $(x_0, g)$ for each environment. In main experiments, for the sake of comparing with RL methods, we use a fixed goal $g$, but note that SLAP and Pure Planning can generalize to new goals (Appendix D.1). At evaluation, we sample 10 held-out tasks per environment and measure (i) success rate and (ii) plan length, which is equivalent to execution time assuming that environment actions are executed at a fixed rate. For RL, we use stable-baselines3 (Raffin et al., 2021) and train each policy for 500,000 steps to obtain the results in Table 1. All PPO policies (shortcut learning, Pure RL, and Hierarchical RL) use the same network architecture and training hyperparameters, except for longer training time and higher entropy coefficient for RL baselines—we tune this to give RL an advantage in exploration. All reported metrics are averaged over 5 random seeds with standard deviations. Additional implementation details and hyperparameters are provided in Appendix B & C.

## 5.1 Results and Discussions

**Empirical Results.** Table 1 summarizes our empirical results. SLAP consistently reduces plan length by large margins compared to Pure Planning. These results highlight the effectiveness of the

| Environment | Approach | Success Rate | Plan Length | Relative Path Length |
|---|---|---|---|---|
| Obstacle 2D | **SLAP** (Ours) | $100\% \pm 0\%$ | $\mathbf{17.6 \pm 1.5}$ | $\downarrow \mathbf{32}\% \pm \mathbf{7}\%$ |
| | Pure Planning | $100\% \pm 0\%$ | $25.9 \pm 1.7$ | $0\%$ |
| | PPO | $0\% \pm 0\%$ | $100.0 \pm 0.0$ (max) | N/A |
| | SAC+HER | $0\% \pm 0\%$ | $100.0 \pm 0.0$ (max) | N/A |
| | Hierarchical RL | $100\% \pm 0\%$ | $25.3 \pm 1.8$ | $\downarrow 2\% \pm 9\%$ |
| | SOL | $100\% \pm 0\%$ | $24.9 \pm 1.2$ | $\downarrow 4\% \pm 8\%$ |
| Obstacle Tower | **SLAP** (Ours) | $100\% \pm 0\%$ | $\mathbf{79.2 \pm 3.2}$ | $\downarrow \mathbf{68}\% \pm \mathbf{2}\%$ |
| | Pure Planning | $100\% \pm 0\%$ | $245.8 \pm 10.4$ | $0\%$ |
| | PPO | $0\% \pm 0\%$ | $500.0 \pm 0.0$ (max) | N/A |
| | SAC+HER | $0\% \pm 0\%$ | $500.0 \pm 0.0$ (max) | N/A |
| | Hierarchical RL | $0\% \pm 0\%$ | $500.0 \pm 0.0$ (max) | N/A |
| | SOL | $0\% \pm 0\%$ | $500.0 \pm 0.0$ (max) | N/A |
| Cluttered Drawer | **SLAP** (Ours) | $100\% \pm 0\%$ | $\mathbf{165.8 \pm 43.6}$ | $\downarrow \mathbf{53}\% \pm \mathbf{14}\%$ |
| | Pure Planning | $100\% \pm 0\%$ | $352.1 \pm 49.5$ | $0\%$ |
| | PPO | $0\% \pm 0\%$ | $500.0 \pm 0.0$ (max) | N/A |
| | SAC+HER | $0\% \pm 0\%$ | $500.0 \pm 0.0$ (max) | N/A |
| | Hierarchical RL | $0\% \pm 0\%$ | $500.0 \pm 0.0$ (max) | N/A |
| | SOL | $0\% \pm 0\%$ | $500.0 \pm 0.0$ (max) | N/A |
| Cleanup Table | **SLAP** (Ours) | $100\% \pm 0\%$ | $\mathbf{115.2 \pm 12.3}$ | $\downarrow \mathbf{73}\% \pm \mathbf{4}\%$ |
| | Pure Planning | $100\% \pm 0\%$ | $431.8 \pm 33.1$ | $0\%$ |
| | PPO | $0\% \pm 0\%$ | $500.0 \pm 0.0$ (max) | N/A |
| | SAC+HER | $0\% \pm 0\%$ | $500.0 \pm 0.0$ (max) | N/A |
| | Hierarchical RL | $0\% \pm 0\%$ | $500.0 \pm 0.0$ (max) | N/A |
| | SOL | $0\% \pm 0\%$ | $500.0 \pm 0.0$ (max) | N/A |

Table 1: **Main Empirical Results.** We report average performance over 10 random seeds with standard deviations. SLAP successfully solves the long-horizon tasks and achieves substantially shorter plans—up to a 73% reduction in plan length—compared to Pure Planning.

shortcuts learned by SLAP (**Q1**). In contrast, pure RL methods struggle to solve these long-horizon tasks, largely due to the sparsity of reward signals, which are received only upon task completion. Even though hierarchical RL methods can mitigate this issue and have additional access to our predefined skills, their high-level controllers struggle to learn the skill selection sequence given the large number of grounded skills in manipulation tasks (**Q2**). For example, SOL needs to deal with 216 grounded skills in Obstacle Tower, compared to 2-3 option policies in the NetHack Learning Environment (Küttler et al., 2020) it was evaluated on in Henaff et al. (2025).

**Training Steps Analysis.** To understand how training time affects performance, we analyze the relationship between shortcut policy training steps and plan lengths (**Q3**). After random rollout pruning, SLAP identifies 11 shortcuts in Obstacle 2D, 92 in Obstacle Tower, 74 in Cluttered Drawer, and 54 in Cleanup Table. As shown in Figure 4, increasing the number of training steps leads to more shortcuts being successfully learned from the fixed set of shortcut candidates and incorporated as graph edges during evaluation. Average plan lengths continue to decrease towards the end of 500,000 training steps as the same shortcut policies become more stable. The marginal benefit varies by environment complexity.

**Generalization Capability Analysis.** We next evaluate the extent to which SLAP can generalize to tasks with new, more, and fewer objects as well as to changes in dynamic properties (**Q4**). We focus on mass and friction, which most strongly affect multi-object interactions in our preliminary tests. SLAP is trained in Obstacle Tower with three stacked obstacles (each with mass 0.5kg and friction coefficient 0.9), and then evaluated on tasks with varying numbers of stacked obstacles and distractor objects scattered on the table using doubled mass and friction at test time. As shown in Figure 5, SLAP maintains short plan lengths even as the number of objects increases, whereas Pure Planning scales poorly with each additional obstacle. These multi-object RL skills ("slap", "wiggle", "wipe")—unlike TAMP methods that assume single-object contact (Billard & Kragic, 2019)—both improve execution efficiency and support generalization over the number of objects. For example, the "slap" shortcut policy shown in Figure 5 only deems a subset of the obstacles relevant but physically affects the entire tower.

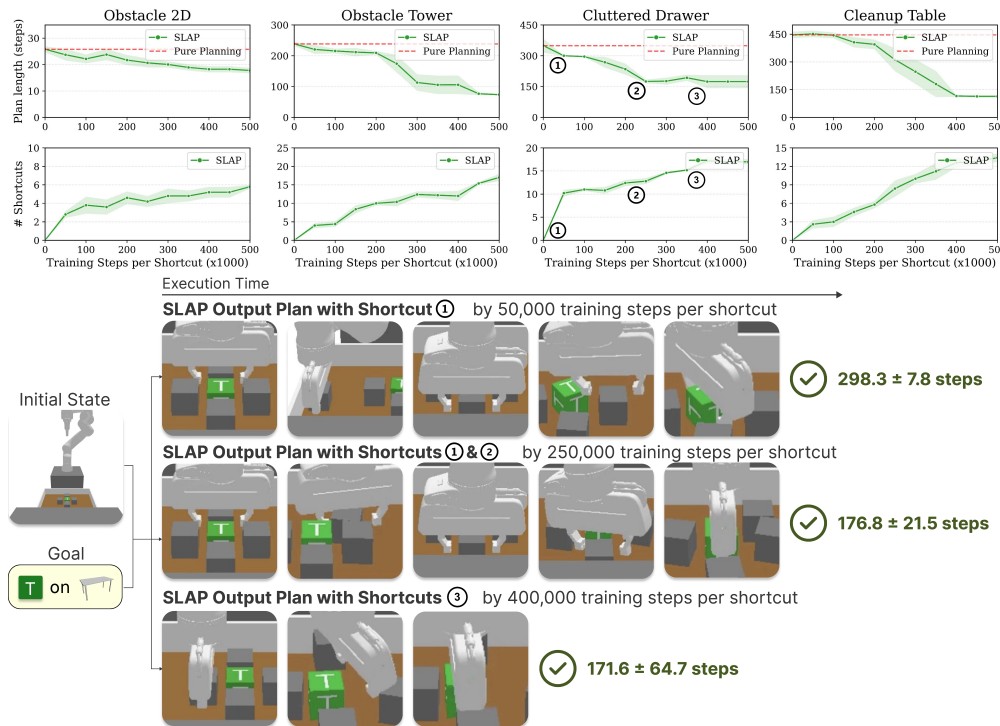

Figure 4: **Training Dynamics**. As the number of training steps increases, more shortcuts are added and the length of the output SLAP plan decreases. In Cluttered Drawer, we visualize SLAP's output plans after different training steps to illustrate which shortcuts are learned and used over time.

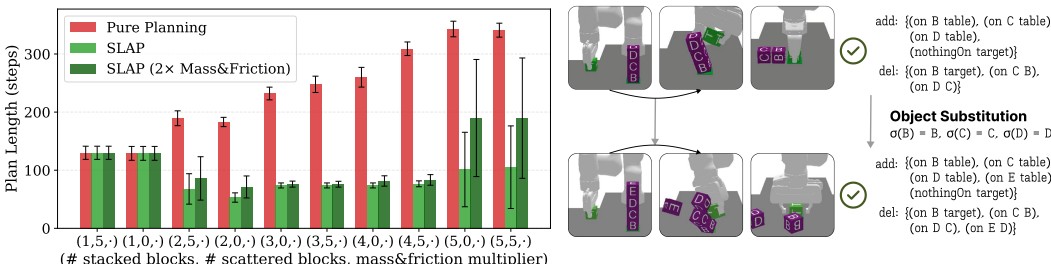

Figure 5: **Generalization Results**. In Obstacle Tower, SLAP is trained on tasks with a stack of 3 obstacles, no distractors. At test time, we are able to generalize to tasks with different numbers of obstacles and distractors, each with different mass and friction, by substituting relevant objects.

**Shortcut Policy Learning Analysis:** SLAP learns separate shortcut policies for each pair of abstract states. While this parallelization makes distributed training easier, it is also possible that training a universal policy (Kaelbling, 1993; Schaul et al., 2015; Eysenbach et al., 2022) could lead to better sample complexity, with learned representations shared across shortcuts (**Q5**). To test this possibility, we compare three shortcut policy learning schemes:

- **Independent**: We train all shortcut policies separately (the default for SLAP).

- **Abstract Subgoals**: We augment observations with a multi-hot encoding of the abstract terminal state for the respective shortcut and train a single shared policy for all shortcuts.

- **Abstract HER**: We use the same multi-hot abstract terminal state encoding as in Abstract Subgoals, but we additionally perform hindsight goal relabeling (as in HER) where goals are now abstract terminal states. During training, if the shortcut policy reaches a different abstract terminal state from the one it was targeting, that data is used to learn about the shortcut that was incidentally achieved. We again use SAC for compatibility with HER.

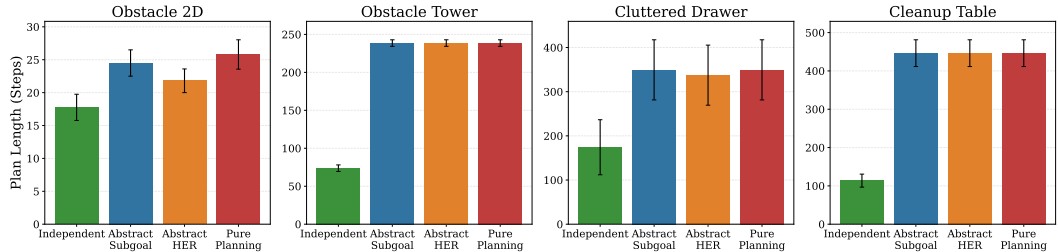

Figure 6: **Shortcut Policy Learning Analysis**. Independent shortcut policy learning consistently finds shorter plans than Abstract Subgoal and Abstract HER.

In Figure 6, we see that Independent consistently outperforms Abstract Subgoals and Abstract HER, particularly in PyBullet environments, despite the opportunity to share representations across shortcuts. We speculate that the poor performance of universal policy learning is due to the fact that shortcuts have varying levels of difficulty for RL. Universal policy learning may implicitly devote resources to learning infeasible shortcuts, where Independent would simply fail to learn on those shortcuts. In future work, we plan to continue exploring shortcut policy learning schemes. We further ablate RL algorithmic choices for shortcut learning in Appendix B.2, with different reward-shaping and exploration strategies (**Q5**).

## 6    DISCUSSION AND CONCLUSION

In this work, we proposed Shortcut Learning for Abstract Planning (SLAP). Our key insight is that the abstract planning graph induced by predefined skills presents an opportunity to learn shortcuts that improve on the execution time of pure planning. In experiments, we showed that the trajectories found by SLAP are better than pure planning in terms of length, and better than pure RL in terms of success rate. We also showed that SLAP can leverage the same relational inductive bias that TAMP uses to solve tasks that feature new, more, fewer objects than those seen during training.

One limitation of SLAP is that it is not able to deviate from the problem decomposition induced by the user-provided options. Future work could consider using the options to instead provide a "soft" problem decomposition that can be further improved by hierarchical RL (Kulkarni et al., 2016; Bacon et al., 2017; Nachum et al., 2018). Another limitation of our work here is that our planner is simple from a TAMP perspective (Section 2). Scaling to very large abstract spaces would benefit from more advanced planning techniques (Garrett et al., 2021). We also made the assumption in this work that the user-provided options are sufficient for solving tasks. Without this assumption, SLAP still applies, but we lose the guarantee of task success. However, in this case, the shortcuts learned by SLAP could improve on the task success rate of pure planning; see Section D.3 for detailed discussion. In addition, because SLAP learns shortcut behaviors that fall outside the set of manually defined options, its execution can be less predictable than traditional TAMP; future work can incorporate safety constraints into shortcut learning. Another direction for future work is combining SLAP with other work that learns abstractions for abstract planning. For example, with demonstration data, we could first learn state abstractions with Silver et al. (2023), action abstractions with Silver et al. (2022), and then leverage our SLAP framework to continue improving the planning efficiency. A final opportunity to extend SLAP is to remove our assumption of access to simulator and, instead, leverage the recent real-to-sim-to-real techniques (Lim et al., 2022; Zhu et al., 2025) to reconstruct approximate simulators from real-world data and learn shortcut policies within those reconstructed environments.

## 7    REPRODUCIBILITY STATEMENT

To support reproducibility, we provide the full source code and models as part of the supplementary materials. All hyperparameters, environment details, and implementation specifics necessary to reproduce the experiments are reported in Appendix B & C.

## 8 ACKNOWLEDGMENTS

We thank Danfei Xu, Chongyi Zheng, and Yixuan Huang for comments on an earlier draft. We would like to give special thanks to Princeton Research Computing[2] for supplying us with the compute that makes our experiments possible.

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

# A APPROACH DETAILS

## A.1 ABSTRACT PLANNING GRAPH

Following previous works (Li et al., 2025; Kumar et al., 2024), the abstract states $s \in \mathcal{S}$ are ground atoms induced from a set of predicates $\psi \in \Psi$. For example, the predicate `On(?o1,?o2)` describes if an object is placed on top of another object. Each predicate is a classifier over the low-level states, with a grounding function that maps continuous state representations into discrete truth values. Every option $a \in \mathcal{A}$ in the experimented environments is equipped with a planning operator $\text{op}^a$ written in the Planning Definition Domain Language (PDDL). Specifically, $\text{op}^a = \langle \text{Var}, \text{Pre}, \text{Eff}^+, \text{Eff}^- \rangle$, where Var is a tuple of object placeholders, and Pre, $\text{Eff}^+$, $\text{Eff}^- \subseteq \Psi$, respectively *preconditions*, *add effects*, and *delete effects*, are each a set of lifted predicates defined with variables in Var. Given an initial state $x_0 \in \mathcal{X}$, we first use the predicates $\Psi$ to obtain the abstract state $s_0$. With the set of operators $\{\text{op}^a, a \in \mathcal{A}\}$, we then build the abstract planning graph in two levels:

**Top Level.** We build the top level of the abstract planning graph using the predefined options, with no simulator required. Starting from $\text{abstract}(x_0)$, we conduct breadth-first search (BFS) until the goal is reached by an abstract state. We expand from each node: (1) ground every operator in the planning domain with all possible combinations of the typed objects, (2) check if any operator's preconditions are satisfied by the current abstract state, and if yes, (3) for every such operator, apply its add and delete effects to obtain the next abstract state, and draw a directed edge to the new node.

At least one abstract state that satisfies the goal will be found after building the top level of the abstract planning graph, since we assume access to sufficiently robust TAMP options in fully-observable and deterministic environments. It is possible that more than one abstract state $\bigcup_{s_g \in \mathcal{S}_g} s_g$ in the existing graph satisfy the goal when BFS terminates. This means that they are at the same depth in the top level of the graph, and the number of high-level steps for the corresponding plans are the same. We argue that SLAP has the advantage of exploring which $s_g$ will benefit from the learned shortcuts the most. For example, in the Obstacle Tower environment, Pure Planning sometimes outputs plans where obstacle blocks are stacked on top of each other after being moved away from the target area. In contrast, SLAP consistently outputs plans with $s_g$ as all obstacle blocks being scattered on the table, because it is only towards this $s_g$ that the most effective "slap" shortcut can be leveraged.

**Bottom Level.** Not all the nodes and edges in the top level can be reached in practice. For example, in the Cluttered Drawer environment, just by grounding the options, the graph includes an abstract state where the robot directly reaches a grasping position around the target object, but this is impossible in practice due to the clutter. To consolidate the abstract planning graph, we start at the root node $x_0$ in the bottom level, conduct BFS to preserve only the feasible parts of the graph, and for each node record the set of low-level states reached by different incoming paths.

The shortest path is found in the bottom level for an execution-time-minimizing solution. For this, we use a path-dependent adaption of Dijkstra's algorithm such that we can re-expand a node if the accumulated edge costs of a new path is lower. This new adaptation turns out to be only slightly more expensive than the original Dijkstra's algorithm. At evaluation, we stop expanding the more expensive branches when the shortest path to goal has already been exposed. We argue that, compared to the number of low-level steps saved at execution, the computational overhead of planning with abstract planning graph augmented by shortcuts is minor. See Appendix D.2 for relevant results.

## A.2 OBJECT SUBSTITUTION FOR SHORTCUT GENERALIZATION

To complement the description in Section 4.4, we provide pseudo-code for the object-substitution mechanism used during SLAP evaluation for shortcut generalization. Algorithm 3 checks whether a learned shortcut can be reused on a different number of new objects by searching for a type-preserving, injective object mapping that preserves the shortcut's add/delete effects.

Our use of symbolic equivalence to guide object substitution is intentional: SLAP reuses a shortcut policy whenever its expected symbolic add/del effects can be matched under a substitution. This design enables generalization of learned shortcuts to different numbers of objects (Section 5.1), different goals (Appendix D.1), and objects with out-of-distribution physical configurations (Section 5.1 & Appendix D.5)—as long as their symbolic effects align.

---

**Algorithm 3:** Object-Substitution for Shortcut Generalization

---

**Input:** $\hat{a}_{\text{train}}$, $\hat{a}_{\text{eval}}$                          `// two shortcut transitions`
**Output:** (`success`, $\sigma$)

**// Construct relevant atom sets**
$\text{Atoms}_{\text{train}} \leftarrow \text{add}(\hat{a}_{\text{train}}) \cup \text{del}(\hat{a}_{\text{train}})$
$\text{Atoms}_{\text{eval}} \leftarrow \text{add}(\hat{a}_{\text{eval}}) \cup \text{del}(\hat{a}_{\text{eval}})$

**// Feasibility checks: predicates and object types**
**if** INFEASIBLEPREDICATES($\text{Atoms}_{\text{train}}$, $\text{Atoms}_{\text{eval}}$) **then**
    ⌊ **return** (`False`, $\emptyset$)
**if** INFEASIBLETYPES($\text{Atoms}_{\text{train}}$, $\text{Atoms}_{\text{eval}}$) **then**
    ⌊ **return** (`False`, $\emptyset$)

**// Search for a type-preserving, injective substitution**
$\sigma \leftarrow \emptyset$
Build candidate sets $C(o)$ for each $o \in \text{rel}(\hat{a}_{\text{train}})$ from objects in $\text{rel}(\hat{a}_{\text{eval}})$ with matching type.
**for** *each $o$ in a fixed ordering of* $\text{rel}(\hat{a}_{train})$ **do**
    choose $o' \in C(o)$ satisfying:
        (i) $o'$ is not already assigned in $\sigma$;
        (ii) for every atom $a \in \text{Atoms}_{\text{train}}$, the atom obtained by substituting each object in $a$
            according to $\sigma \cup \{o \mapsto o'\}$ also appears in $\text{Atoms}_{\text{eval}}$.
    **if** *no such $o'$ exists* **then**
        ⌊ **return** (`False`, $\emptyset$)                         `// early prune`
    $\sigma \leftarrow \sigma \cup \{o \mapsto o'\}$
**return** (`True`, $\sigma$)

---

## B    ENVIRONMENT AND EXPERIMENT DETAILS

In this section, we provide the detailed operators, options, and predicates for each environment, as well as experiment settings. For more details, please refer to our open-sourced code.

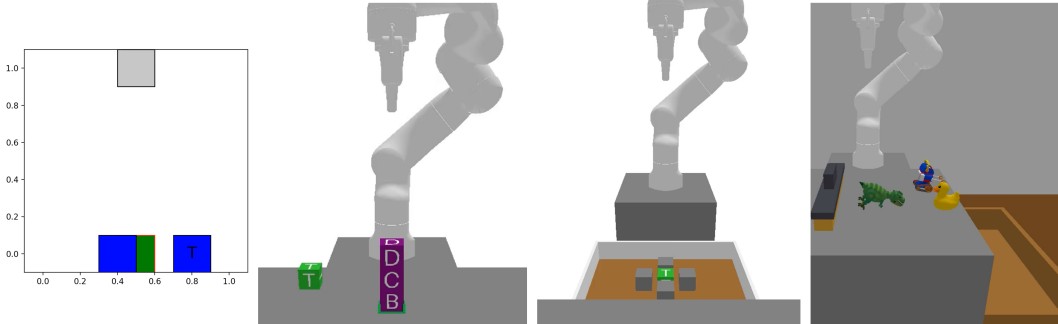

Figure 7: **Environment Visualization**. From left to right are our four environments featuring long horizons, sparse rewards, and various physical interactions: Obstacle 2D, Obstacle Tower, Cluttered Drawer, Cleanup Table.

**Obstacle 2D:**

- *Operators & Options*: `Pick`, `Place`, `PickFromTarget`, and `PlaceInTarget`.
- *Predicates*: `IsBlock(?o)`, `IsSurface(?o)`, `IsRobot(?o)`, `On(?o1, ?o2)`, `Overlap(?o1, ?o2)`, `Holding(?o1, ?o2)`, `GripperEmpty(?o)`, `Clear(?o)`, `IsTarget(?o)`, `NotIsTarget(?o)`.
- *Task Description*: There is a 1-by-1 target area in the middle of the bottom line, the exact size of one block. The agent controls a gripper that can interact with the blocks. The goal of the task is to place a target block in the target area. The other scattered blocks are either

blocking the target area in their initial positions (i.e. obstacle blocks) or somewhere else on the bottom line (i.e. irrelevant blocks). Therefore, in order to reach the goal, the agent would have to interact with the obstacle blocks to clear the target area. The initial positions of the blocks are randomized while guaranteeing that at least one of the blocks will partially block the target area to make the task harder. During generalization test for different number of objects, we add one additional block on the bottom line given the limited space.

- *Typical Shortcut from SLAP*: Our algorithms learns a shortcut policy that "pushes" the obstacle in the target region while holding the target block.

- *Experiment Setup*: We sample 10 tasks for this environment. For each sampled task, we randomly roll out $N_{\text{rollout}} = 1000$ episodes, with $T_{\text{rollout}} = 100$ max steps per episode. In the random rollouts, we used a threshold of $K_{\text{rollout}} = 1$ to identify promising shortcuts. From all the tasks and episodes, SLAP found 11 shortcuts, with 40 scenarios in total. During training, we implement PPO algorithm with a batch size of 16, a learning rate of 3e-4, and an entropy coefficient of 0.01 for each of the shortcut policy learning. The shortcut policies are trained for 1000 episodes with 50 maximum steps per episode to obtain the shortest output plans we have observed.

**Obstacle Tower:**

- *Operators & Options*: `Pick`, `Place`, `Stack`, `Unstack`, `PickFromTarget`, `PlaceInTarget`.

- *Predicates*: `IsBlock(?o), IsSurface(?o), IsRobot(?o), IsMovable(?o), NotIsMovable(?o), On(?o1, ?o2), NothingOn(?o), Holding(?o1, ?o2), NotHolding(?o1, ?o2), GripperEmpty(?o), IsTarget(?o), NotIsTarget(?o).`

- *Task Description*: This environment is a 3D PyBullet adaptation of the `Blocks2D` environment with more complexities. It has a table, Franka Emika Panda 7-DOF robot, and some lettered blocks on the table. A small area is marked as target area on the table, and one of the blocks is marked with letter 'T' to be the target block. As for the other blocks, some are stacked in the target area and blocking it almost fully, while others are scattered elsewhere on the table. Similar to `Blocks2D`, the goal is to place block T in the target area, but this would require moving the other blocks away from the target area first. During generalization test for different number of objects, we adjust the number of stacked blocks in the target area and randomly scatter additional blocks on the table.

- *Typical Shortcut from SLAP*: Our algorithm learned a shortcut policy that "slaps" the block tower on the target region to make it clear. The policy can be instantiated after the target block is picked up.

- *Experiment Setup*: We sample 10 tasks for this environment. For each sampled task, we randomly roll out $N_{\text{rollout}} = 100$ episodes, with $T_{\text{rollout}} = 300$ max steps per episode. In the random rollouts, we used a threshold of $K_{\text{rollout}} = 5$ to identify promising shortcuts. From all the tasks and episodes, SLAP found 92 shortcuts, with 1070 scenarios in total. During training, we implement PPO algorithm with a batch size of 16, a learning rate of 3e-4, and an entropy coefficient of 0.01 for each of the shortcut policy learning. The shortcut policies are trained for 3000 episodes with 100 maximum steps per episode to obtain the shortest output plans we have observed.

**Cluttered Drawer:**

- *Operators & Options*: `Reach`, `GraspFrontBack`, `GraspLeftRight`, `GraspFullClear`, `GraspNonTarget`, `PlaceTarget`, `PlaceFrontBlock`, `PlaceBackBlock`, `PlaceLeftBlock`, `PlaceRightBlock`.

- *Predicates*: `IsBlock(?o), IsTable(?o), IsDrawer(?o), IsRobot(?o), IsMovable(?o), NotIsMovable(?o), ReadyPick(?o), NotReadyPick(?o), On(?o1, ?o2), Holding(?o1, ?o2), NotHolding(?o1, ?o2), GripperEmpty(?o), IsTargetBlock(?o), NotIsTargetBlock(?o), BlockingLeft(?o1, ?o2), BlockingRight(?o1, ?o2), BlockingFront(?o1, ?o2),`

```
BlockingBack(?o1, ?o2), LeftClear(?o), RightClear(?o),
FrontClear(?o), BackClear(?o), HandReadyPick(?o).
```

- *Task Description*: In this environment, a Franka Emika Panda 7-DOF robot aims to grasp a target block inside a cluttered drawer and place it on the table. Initially, there are a number of obstacles that make the target block not graspable with given motion skills. Therefore, the abstract planner will try to reach, grasp, and place each of these obstacles to make at least two sides of the target block clear (graspable). During training, there are four obstacles blocking the right, left, front, and back sides of the target block, respectively. During generalization test for different number of objects, we randomly scatter additional obstacles in the drawer.

- *Typical Shortcut from SLAP*: Our algorithm learns to "wiggle" the robot hand around the target block with the fingers open, such that its sides become clear. This shortcut policy is used with the given skills during planning.

- *Experiment Setup*: We sample 10 tasks for this environment. For each sampled task, we randomly roll out $N_{\text{rollout}} = 100$ episodes, with $T_{\text{rollout}} = 300$ max steps per episode. In the random rollouts, we used a threshold of $K_{\text{rollout}} = 5$ to identify promising short cuts. From all the tasks and episodes, SLAP found 74 shortcuts, with 1012 scenarios in total. During training, we implement PPO algorithm with a batch size of 16, a learning rate of 3e-4, and an entropy coefficient of 0.01 for each of the shortcut policy learning. The shortcut policies are trained for 1500 episodes with 100 maximum steps per episode to obtain the shortest output plans we have observed.

**Cleanup Table:**

- *Operators & Options*: `Reach, Grasp, Lift, Drop,`

- *Predicates*: `IsBlock(?o), IsTable(?o), IsDrawer(?o), IsRobot(?o), IsMovable(?o), NotIsMovable(?o), ReadyPick(?o), NotReadyPick(?o), On(?o1, ?o2), Holding(?o1, ?o2), NotHolding(?o1, ?o2), GripperEmpty(?o), HandReadyPick(?o), AboveEverything(?o), NotAboveEverything(?o).`

- *Task Description*: In this environment, a Franka Emika Panda 7-DOF robot aims to move all the objects on the table to a storage bin. These objects include three toys (duck toy, dinosaur toy, and robot toy) and a small wiper. All of them have highly irregular and realistic mesh shapes imported from Objaverse Deitke et al. (2023). The abstract planner plans to pick each irregular object up and drops it into the bin. The toys are randomly scattered on the table in each episode; at initial placements, we check collisions using slightly enlarged bounding spheres to compensate for mesh inaccuracies introduced by downscaling (to have realistic toy sizes relative to the robot). During generalization test for different number of objects, we randomly scatter fewer or more Objaverse toy objects on the table.

- *Typical Shortcut from SLAP*: Our algorithm learns to picks up the small wiper tool first and slowly "sweeps" at a certain height and an appropriate direction such that all the toy objects are gathered into the storage bin at once without falling off the small table in the middle of the process. This shortcut policy is instantiated after the wiper is picked up.

- *Experiment Setup*: We sample 10 tasks for this environment. For each sampled task, we randomly roll out $N_{\text{rollout}} = 100$ episodes, with $T_{\text{rollout}} = 300$ max steps per episode. In the random rollouts, we used a threshold of $K_{\text{rollout}} = 5$ to identify promising short cuts. From all the tasks and episodes, SLAP found 54 shortcuts, with 770 scenarios in total. During training, we implement PPO algorithm with a batch size of 16, a learning rate of 3e-4, and an entropy coefficient of 0.01 for each of the shortcut policy learning. The shortcut policies are trained for 3500 episodes with 100 maximum steps per episode to obtain the shortest output plans we have observed.

The hyperparameters for RL shortcut learning are exactly the same for all the environments, since shortcut policies do not require any hyperparameter tuning. The hyperparameters $N_{\text{rollout}}, T_{\text{rollout}}$, and $K_{\text{rollout}}$ for random rollouts pruning are consistent across all PyBullet environments but are adjusted for the Obstacle 2D environment for its significantly lower complexity.

## B.1 ABLATIONS ON RANDOM ROLLOUTS PRUNING

For our experiments on the three PyBullet environments, we use a consistent set of hyperparameters for random rollouts pruning, with a ratio of $K_{\text{rollout}}/N_{\text{rollout}} = 5/100 = 5\%$ for selecting shortcuts to learn. Ablation results of this ratio are shown in Table 2. Useful shortcuts are pruned significantly when the threshold is around 20%. In general, this threshold provides a mechanism to trade off training time and plan length.

| Ratio $K_{rollout}/N_{rollout}$ | Success Rate | Plan Length |
|:---:|:---:|:---:|
| 5% | $100\% \pm 0\%$ | $178.2 \pm 58.1$ |
| 10% | $100\% \pm 0\%$ | $167.0 \pm 46.1$ |
| 15% | $100\% \pm 0\%$ | $185.0 \pm 54.4$ |
| 20% | $100\% \pm 0\%$ | $209.0 \pm 79.1$ |
| 25% | $100\% \pm 0\%$ | $202.0 \pm 81.6$ |
| 30% | $100\% \pm 0\%$ | $333.0 \pm 36.4$ |
| 35% | $100\% \pm 0\%$ | $349.0 \pm 74.0$ |

Table 2: **Ablations on $K_{\text{rollout}}/N_{\text{rollout}}$.** We varied such ratio in the experiments on Cluttered Drawer and report average performance over 3 random seeds.

Our random-rollout pruning mechanism is simple, but it is task-agnostic and generally applicable. It is also surprisingly effective: for example, in the Cluttered Drawer environment, 98.70% of all possible shortcuts are pruned. To further validate the technique, we conducted an additional experiment in Cluttered Drawer where there are 5623 pruned shortcuts. We randomly selected 200 of the pruned shortcuts to train with RL on one seed. The overall shortcut training success rates are $3.64\% \pm 12.15\%$, with 9 of them successfully added at evaluation time. The number of execution steps are comparable to that of Pure Planning, except in one evaluation episode where placing and lifting a non-target object are replaced by a shortcut of dropping it with a "jittering" movement of the robot arm. Overall, these results confirm that these shortcuts can be pruned with little impact to test-time performance.

## B.2 ABLATIONS ON SHORTCUT LEARNING POLICIES

In Section 4.2, we argued that any continuous-state, continuous-action RL algorithm can serve as the backbone for shortcut learning. To substantiate the claim, we present ablations on the Cleanup Table environment using three different backbone algorithms: proximal policy learning (PPO) (Schulman et al., 2017), soft actor-critic (SAC) (Haarnoja et al., 2018), or SAC with hindsight experience replay (SAC+HER) (Andrychowicz et al., 2017).

| Environment | Approach | Success Rate | Plan Length | Relative Path Length |
|:---|:---|:---:|:---:|:---:|
| | **SLAP (PPO)** | $100\% \pm 0\%$ | $\mathbf{113.7 \pm 17.0}$ | $\downarrow \mathbf{74.5\% \pm 4.5\%}$ |
| | SLAP (SAC) | $100\% \pm 0\%$ | $160.2 \pm 19.4$ | $\downarrow 64.1\% \pm 5.0\%$ |
| Cleanup Table | SLAP (SAC+HER) | $100\% \pm 0\%$ | $131.0 \pm 19.5$ | $\downarrow 70.6\% \pm 5.2\%$ |
| | Pure Planning | $100\% \pm 0\%$ | $446.3 \pm 34.9$ | $0\%$ |

Table 3: **Results of Different Shortcut RL Algorithms on Cleanup Table.** We report average performance over 5 random seeds with standard deviations. Different RL algorithms eventually converge to similar results on shortcuts' execution-time efficiency and overall plan lengths. We observe slightly better performance with PPO as the shortcut policy backbone, consistent with our choice for the main results reported in the paper.

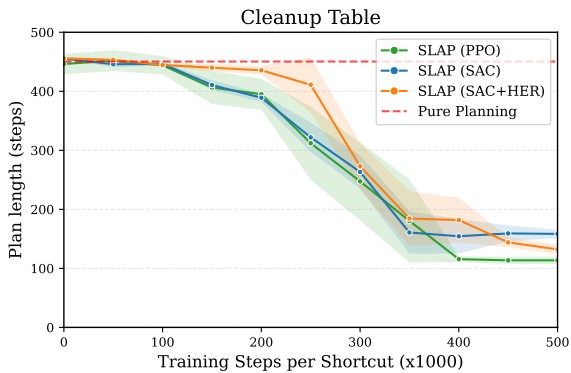

Figure 8: Training Dynamics of Different Shortcut RL Algorithms on Cleanup Table.

## C   BASELINES DETAILS

In this section, we describe the implementation details of all the baselines we have covered in the paper, including the baselines presented in the main results and the ablations used for shortcut policy learning analysis.

**Pure Planning.**   Pure Planning uses the same abstract planner as SLAP but without any learned shortcuts or low-level policies. All actions are executed through predefined skills. The planner operates with the same planning horizon as SLAP to ensure fair comparison. This baseline represents the performance of planning approaches without learning components.

**Pure RL (PPO).**   For the pure RL baseline using PPO, we train policies directly on the full task $(x_0, g)$ with reward functions that penalize execution time through step penalties (and an optional positive bonus reward for achieving the goal). The PPO implementation uses batch size of 16, learning rate of 3e-4, 10 epochs, discount factor of 0.99, and entropy coefficient of 0.05. Compared to our approach SLAP, we spent more time tuning the hyperparameters of the pure RL baselines to ensure fairness. The reported hyperparameters is the setting where we observed a nonzero training success rate for Obstacle 2D environment (0.14% average training success rate). Training is conducted for 1,000,000 total steps with episode lengths matching the environment-specific maximums. Network architectures consist of 2-layer MLPs with 64 hidden units per layer and tanh activations for both policy and value networks.

**Pure RL (SAC+HER).**   Given the sparse reward nature of our environments, we implement a baseline using Soft Actor-Critic with Hindsight Experience Replay. The SAC component uses learning rates of 3e-4 for actor, critic, and temperature networks, with a replay buffer size of 1,000,000, batch size of 16, and target smoothing coefficient of 0.005. The networks are 2-layer MLPs with 256 units per layer. HER is configured with a "future" goal selection strategy and replay ratio of 0.8. Training runs for 1,000,000 steps with an initial exploration phase of 1000 episodes. The reward functions still consist of step penalties; the policies observe a non-negative (or an optional positive bonus reward) reward if they reach a state that is within a distance of 0.01 compared to the goal. This distance-checking is possible because the observations are object-centric, so we only need to extract partial observations that are directly relevant to the task goal.

**Hierarchical RL (PPO).**   The hierarchical RL baseline outputs a combined action vector of low-level controls and skill activations. It is able to complete the tasks in the Obstacle 2D environment after we increased the entropy coefficient to 0.05 for more exploration. However, it fails to solve any of the more complicated PyBullet tasks after we conducted systematic scans of hyperparameters. For the reported experimental results, we use the same model architecture and hyperparameters as the Pure RL (PPO) baseline: learning rate of 3e-4, batch size of 16, 10 epochs per update, discount factor of 0.99, and entropy coefficient of 0.05. Training runs for 1,000,000 steps.

**SOL.** The original SOL algorithm jointly learns controller and option policies via the given intrinsic rewards. We made several modifications to the implementation and presented additional inputs to SOL to make our long-horizon tasks easier for it to learn. First, our SOL assumes access to all the predefined skills grounded with different combinations of typed objects. When a predefined skill is chosen, it is executed until completion and can indicate early skill termination. If the controller calls a predefined skill whose preconditions are not satisfied by the current atoms, the skill outputs no-op actions and returns control to the meta-controller. This is the same during evaluation, we call the controller again instead of reporting errors on the infeasible operators selected in order to reduce task difficulty. Our SOL baseline also assumes access to the shortcut data from the abstract planning graphs to better leverage the hierarchical structure in the same way as SLAP. The intrinsic rewards for shortcut policy learning in SOL are also the same as SLAP – goal-based sparse rewards upon shortcut completion. Furthermore, within the SOL algorithm, we removed the controller penalty: it is easy to select grounded skills with unsatisfied preconditions, and the accumulated penalties often overwhelm the sparse task completion signal and prevent learning. Training is conducted for 50,000,000 total steps with episode lengths twice the environment-specific maximums. We use PPO with learning rate 3e-4, discount factor 0.995, GAE 0.98, exploration coefficient 0.01 (they used 0.0001 for PointMaze environments), with each skill option executing for up to 100 steps.

The two baselines below only differ from SLAP in the RL architecture for learning shortcut policies:

**Abstract Subgoals.** This baseline directly augments the raw environment observations with a multi-hot encoding of the abstract terminal state of the corresponding shortcut, where each atom is mapped to a fixed index in the context vector. A single shared PPO policy is trained across all shortcuts using these augmented observations. We use the same RL hyperparameters for Abstract Subgoals as SLAP (see Appendix B).

**Abstract HER.** This baseline shares the same SAC hyperparameters as the pure RL (SAC+HER) baseline described above. However, instead of sampling goals from training trajectories in the goal relabeling stage as in standard HER, we use a custom NodeBasedHER buffer that samples goals from our planning graph's abstract states. This is equivalent to training a goal-conditioned RL policy on multiple shortcuts, and we limit the pool we sample from to terminal abstract states of promising shortcuts. The goals, same as the abstract subgoals baseline, are represented as multi-hot encodings. The hyperparameters of NodeBasedHER's replay buffer differs from the Pure RL (SAC+HER) baseline with a smaller replay buffer size of 1000 and a larger replay ratio of 0.95. These adjustments are made such that it learns all the promising shortcuts more equally at the same time.

# D  ADDITIONAL EXPERIMENTS

## D.1  GENERALIZATION OVER GOALS

In Section 5.1, we have discussed SLAP's generalization capabilities to tasks with different numbers of objects. Here, we present additional results on SLAP's ability to generalize to new tasks with different goals. As mentioned in Section 4.3, generalization over goals is realized by leveraging the abstract planning graph; different goals correspond to different sets of nodes in the graph. To generalize to a task with a new goal unseen during training, we simply need to find the shortest path to one of the abstract states that satisfy the new goal.

In Figure 9, we see that on average SLAP finds shorter plans than Pure Planning for new tasks with different goals. The large standard deviations are due to the random sampling of abstract goals. An example is shown for the Obstacle Tower environment. The "slap" shortcut is learned during training to achieve the goal of placing the target block in the target area. But in evaluation time, with a new goal of stacking the blocks in reverse order, SLAP is able to use the same shortcut to slap all the blocks on table, and re-stack the blocks directly afterwards.

## D.2  TIME EFFICIENCY

In this section, we include results on the time efficiency of SLAP at test time: How much computational overhead does the abstract planning graph augmented with shortcuts introduce? In Table 1

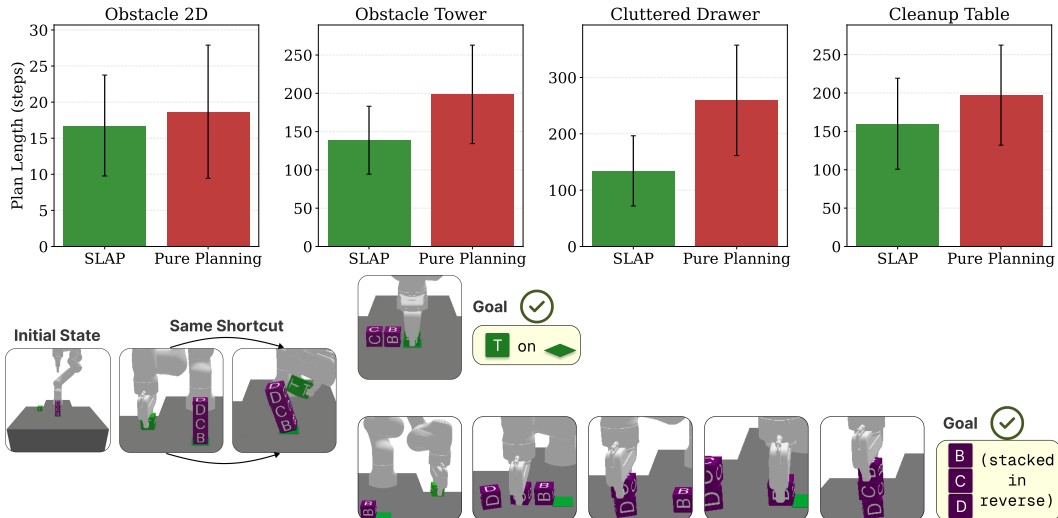

Figure 9: **Generalization to New Tasks**. Results of average plan lengths on 15 tasks with randomly sampled goals in evaluation for each environment. With a fixed set of trained shortcuts, SLAP finds shorter plans on average for these new tasks.

we have shown the proportion of execution time we can save for the robot at deployment. Here we present realistic estimates on the overall time efficiency considering both planning time and execution time at real-world deployment.

| Environment | Approach | Compute time during planning (seconds) | Execution steps | Per-step execution time to choose SLAP over Pure Planning (seconds) |
|---|---|---|---|---|
| Obstacle Tower | SLAP (Ours) | **15.2 ± 1.2** | **73.8 ± 4.3** | Any |
| | Pure Planning | 18.4 ± 0.2 | 238.6 ± 12.8 | |
| Cluttered Drawer | SLAP (Ours) | 275.2 ± 8.0 | **174.2 ± 62.3** | 1.2 |
| | Pure Planning | **65.5 ± 5.2** | 349.4 ± 68.1 | |
| Cleanup Table | SLAP (Ours) | 201.0 ± 3.9 | **113.7 ± 17.0** | 0.5 |
| | Pure Planning | **34.0 ± 5.8** | 446.3 ± 34.9 | |

Table 4: **Time efficiency**. Comparative results of SLAP's and Pure Planning's planning time and execution steps at evaluation. While SLAP incurs slightly higher planning time than Pure Planning in Cluttered Drawer and Cleanup Table, the significant reduction in execution steps more than compensates for this cost under realistic per-step execution time.

In Table 4, we record the clock time of SLAP's planning phase at evaluation on the three PyBullet environments that we have, as PyBullet (Coumans & Bai, 2016) simulates the robot-object interaction physics of our manipulation tasks well and provides a reliable proxy for real-world deployment. Based on the compute time during planning and the number of execution steps SLAP improves compared to Pure Planning, we give lower bounds on the per-step execution time where the users would prefer SLAP over Pure Planning just for time efficiency reasons. In Obstacle Tower environment, we can see that SLAP has lower planning time and execution steps at evaluation, so SLAP is preferable for use regardless of how long each execution step takes in reality. As for Cluttered Drawer and Cleanup Table environments, the lower bounds are 1.20 and 0.50 seconds respectively – values that are within the typical range of per-step execution time for many real-world robots.

Note that we are not using any heuristics to accelerate graph search for SLAP, whereas Pure Planning uses the Fast-Forward heuristic (Hoffmann, 2001) with greedy best-first search to reduce planning time. Heuristics can be integrated with graph search in SLAP to further boost its time efficiency, which we are actively working towards.

### D.3   SLAP FOR ABSTRACT PLANNERS WITH SUBOPTIMAL ABSTRACTIONS

In SLAP, we define shortcuts based on the hierarchical structure of the abstract planning graph induced by the predefined skills. We made the assumption that these user-provided abstractions are sufficiently robust to generate solutions for goals in our task distribution (see assumptions in Section 3) such that SLAP can be applied on top of the planners to further improve their execution time without losing completeness guarantees. However, for complicated real-world robotics problems the abstractions can be suboptimal. We are interested in such cases to see if SLAP's performance will degrade and to what extent.

We modify the Cluttered Drawer environment to test SLAP's performance when the grounding functions of predicates are noisy and imprecise. In particular, the Cluttered Drawer domain includes several predicates (BlockingLeft(?o1, ?o2), BlockingRight(?o1, ?o2), BlockingFront(?o1, ?o2), BlockingBack(?o1, ?o2)) that reflect whether the robot can directly grasp the target object from the cluttered drawer without manipulating the surrounding objects first. The thresholds for such grounding functions are very important, and we add noises to the thresholds for the perceiver at every step to mirror realistic scenarios where we are planning with perception and localization systems that have prediction errors.

In the original implementation, the "blocking" predicate is classified to be true in one direction if the distance between the two objects in that direction is less than the width $w$ of an object. For each set of the experiments below, we define a range for such threshold $[c_1 \cdot w, c_2 \cdot w]$ and randomly sample a threshold for the grounding function at each step. At test time, to handle suboptimal abstractions at the abstract level, we replan on each time step (following previous works like (Yoon et al., 2007)). For fair comparison, we extend Pure Planning to replan as well.

| Environment | Approach | Success Rate | Plan Length | Relative Path Length |
|---|---|---|---|---|
| Cluttered Drawer $[w, 2w]$ | SLAP (Ours) | $0\% \pm 0\%$ | $500.0 \pm 0.0$ (max) | N/A |
| | Pure Planning | $\mathbf{63\% \pm 9\%}$ | $\mathbf{403.5 \pm 77.5}$ | $0\%$ |
| Cluttered Drawer $[w, 1.5w]$ | SLAP (Ours) | $98\% \pm 1\%$ | $\mathbf{195.6 \pm 56.0}$ | $\downarrow \mathbf{45\% \pm 18\%}$ |
| | Pure Planning | $\mathbf{100 \pm 0\%}$ | $358.3 \pm 52.4$ | $0\%$ |
| Cluttered Drawer (Optimal) | SLAP (Ours) | $100\% \pm 0\%$ | $\mathbf{174.2 \pm 62.3}$ | $\downarrow \mathbf{50\% \pm 20\%}$ |
| | Pure Planning | $100\% \pm 0\%$ | $349.4 \pm 68.1$ | $0\%$ |
| Cluttered Drawer $[0.75w, w]$ | SLAP (Ours) | $100\% \pm 0\%$ | $\mathbf{168.0 \pm 45.2}$ | $\downarrow \mathbf{53\% \pm 16\%}$ |
| | Pure Planning | $100\% \pm 0\%$ | $359.4 \pm 56.9$ | $0\%$ |
| Cluttered Drawer $[0.5w, w]$ | SLAP (Ours) | $\mathbf{92\% \pm 5\%}$ | $\mathbf{204.7 \pm 72.1}$ | $\downarrow \mathbf{55\% \pm 17\%}$ |
| | Pure Planning | $54\% \pm 11\%$ | $449.8 \pm 32.1$ | $0\%$ |

Table 5: **Results on variants of Cluttered Drawer with suboptimal abstractions.** We report average performance over 5 random seeds with standard deviations. Abstractions have to shift significantly from the "optimal" before SLAP's performance degrades in comparison to Pure Planning.

From Table 5, when the thresholds for grounding functions are sampled from $[0.5w, w]$, SLAP applied on top of suboptimal abstractions even improves upon the success rates of Pure Planning. Some learned RL shortcuts connect from noisy abstract states with misclassified "blocking" predicates to states that lead to the goal. In comparison, without the flexibility of RL, Pure Planning would replan and be completely misguided by the abstractions.

However, SLAP's performance degrades substantially when the suboptimal abstractions are too noisy. When the thresholds are sampled from $[w, 2w]$, whether the target object is being blocked solely depends on the sampled threshold at the current step, since the probability that a surrounding object is moved to a distance of $2w$ away from the target object is low. In this case, the abstract planning graph can no longer provide a good structure to define helpful shortcuts for RL to learn. Therefore, SLAP's success rate goes to 0%.

## D.4  SLAP in Stochastic, Partially Observable Environments

As mentioned in Section 3, we focus on applications to fully-observable and deterministic environments, aligned with the scope of most TAMP methods (see survey Garrett et al. (2021)). However, we are also interested in SLAP's performance in environments with looser restrictions.

We first introduce a stochastic variant of the Obstacle Tower environment that features noisy actions (1% std Gaussian noise), object physics (random variations in 10% size, 20% mass and friction), stack alignment (1% position noise and 10% rotation noise of each block in the stack), and random dropping (with 1% probability if any object is held). We train shortcut policies with the same setup as in Section 5.1. At test time, we replan on each time step, same as Pure Planning. Results over 5 seeds are shown in Table 6. Similar to the main results in Table 1, SLAP outperforms Pure Planning and RL in terms of plan length. Perhaps surprisingly, in this stochastic setting, SLAP also outperforms Pure Planning in terms of success rate. This is because the shortcuts learned with RL are better able to handle stochasticity than the user-provided options. Qualitatively, instead of relying solely on the held object to push the obstacle stack, the robot bends lower and uses its arm to push.

We also introduce a partially observable variant of Obstacle Tower where the top block of the obstacle stack is occluded and therefore absent from the simulator used for planning. We use pre-trained policies to test generalizability and robustness and again use replanning at test time. The results in Table 6 show that Pure Planning consistently fails; qualitatively, it tries to directly grasp the second block and gets stuck in collision. SLAP attains a 78% success rate (compared to 2% for Pure Planning) using a "slap" shortcut that is similar to the object-based generalization results, but importantly, the planner and shortcut policy do not have knowledge of the occluded block.

| Environment | Approach | Success Rate | Plan Length | Relative Path Length |
|---|---|---|---|---|
| Obstacle Tower (stochastic) | **SLAP** (Ours) | **92% $\pm$ 4%** | **119.7 $\pm$ 103.6** | **$\downarrow$ 59% $\pm$ 36%** |
| | Pure Planning | 84% $\pm$ 6% | 293.6 $\pm$ 58.4 | 0% |
| | PPO | 0% $\pm$ 0% | 500.0 $\pm$ 0.0 (max) | N/A |
| | SAC+HER | 0% $\pm$ 0% | 500.0 $\pm$ 0.0 (max) | N/A |
| | Hierarchical RL | 0% $\pm$ 0% | 500.0 $\pm$ 0.0 (max) | N/A |
| Obstacle Tower (partially observable) | **SLAP** (Ours) | **78% $\pm$ 8%** | **178.0 $\pm$ 182.0** | **$\downarrow$ 64% $\pm$ 37%** |
| | Pure Planning | 2% $\pm$ 2% | 493.7 $\pm$ 44.4 | 0% |
| | PPO | 0% $\pm$ 0% | 500.0 $\pm$ 0.0 (max) | N/A |
| | SAC+HER | 0% $\pm$ 0% | 500.0 $\pm$ 0.0 (max) | N/A |
| | Hierarchical RL | 0% $\pm$ 0% | 500.0 $\pm$ 0.0 (max) | N/A |

Table 6: **Results on variants of Obstacle Tower with looser environment assumptions.** We report average performance over 5 random seeds with standard deviations. The shortcuts learned with RL are better at handling stochasticity. And because shortcuts like "slap" involve the robot's interactions with multiple objects at once, the task success rates are much higher compare to Pure Planning when a subset of the objects is fully occluded.

## D.5  SLAP Under Out-of-Distribution Physical Configurations

In many robotic settings, the physical properties of objects at test time (e.g., mass, friction, size, or contact noise) may differ from those seen during training. To evaluate SLAP's robustness under such out-of-distribution physical configurations, we compare: (i) SLAP trained on the standard, deterministic Obstacle Tower environment and evaluated under heavily perturbed physics, and (ii) SLAP trained directly on the perturbed environment.

Our perturbed configuration introduces noisy actions (2% Gaussian), random object physics (10% size variation, 30% mass/friction variation), stack alignment noise (1% position, 10% rotation), and random dropping (2% probability whenever an object is held). These perturbations significantly affect multi-object interactions and present a challenging test of robustness.

These results show that SLAP's learned shortcuts exhibit strong robustness to out-of-distribution physical perturbations even without retraining, and that training directly on perturbed physics further improves performance. This provides guidance for practitioners: SLAP can generalize effectively

| Environment | Approach | Success Rate | Plan Length | Relative Path Length |
|---|---|---|---|---|
| Obstacle Tower (perturbed) | **SLAP (Ours)** | **78% ± 9%** | **151.3 ± 115.8** | ↓ **58% ± 32%** |
| | Pure Planning | 65% ± 10% | 357.5 ± 54.2 | 0% |

Table 7: **SLAP trained on deterministic physics.** SLAP can handle such physical perturbations better than Pure Planning. Results are reported across 5 seeds.

| Environment | Approach | Success Rate | Plan Length | Relative Path Length |
|---|---|---|---|---|
| Obstacle Tower (perturbed) | **SLAP (Ours)** | **86% ± 7%** | **133.4 ± 98.2** | ↓ **63% ± 26%** |
| | Pure Planning | 65% ± 10% | 357.5 ± 54.2 | 0% |

Table 8: **SLAP trained on perturbed physics.** As expected, performance improves further, but even in this challenging setting SLAP still exhibits failures due to large perturbations.

across moderate changes in physical properties, but for significantly altered dynamics it may be beneficial to train a new set of shortcuts.

