# OpenReview forum: "SLAP: Shortcut Learning for Abstract Planning"
_ICLR.cc/2026/Conference — ICLR 2026 Poster_

### Official Review · Reviewer_sKAL · 2025-10-16

**Soundness:** 2
**Presentation:** 3
**Contribution:** 2
**Rating:** 4
**Confidence:** 3

**Summary:**

This paper introduces Shortcut Learning for Abstract Planning (SLAP), a novel framework that enhances robotic problem-solving by synergizing traditional Task and Motion Planning (TAMP) with Reinforcement Learning (RL). Addressing the limitation that TAMP relies on pre-programmed and often inefficient skills, SLAP autonomously discovers more effective behaviors by analyzing the high-level structure of an abstract plan to identify potential "shortcuts." It then uses RL to train new, dynamic, low-level skills (such as "slapping" or "wiping" obstacles) to bridge these gaps in the plan. These learned shortcut policies are integrated back into the planner, enabling it to find significantly shorter and more efficient solutions. The key contributions are a dramatic reduction in plan length by over 50%, a high success rate on complex, long-horizon tasks where pure RL methods fail, and strong generalization to new tasks and objects not seen during training.

**Strengths:**

- Figure 1 is very well made and clear
- The explanation of how the work fits into the existing literature is very good, even for someone unfamiliar with the relevant literature
- The results seem pretty amazing in terms of improvement over baselines

**Weaknesses:**

- Figure 2 and 3 are pretty confusing in my opinion and it is not clear just from looking at the figure what the grey and black balls are supposed to represesent -- this should be made clearer so that it is understandable just from looking at the figure
- Some analysis of sample/computational efficiency should be provided in the main text and results (how many extra samples, how much extra computation, how much extra training time is required to learn these shortcuts). There may be an appendix related to this but I believe at least a summary should be included in the main text so that the reader can know if potentially this improvement in performance is just from giving extra data to the learning algorithm through the shortcut learning policies
- Some discussion of how this algorithm might be used when there is not access to a simulator (or if it just not possible in that case) should also be included, or what additional pieces would be needed in the case of no simulator
- I am unsure of the novelty of the approach. It seems like SLAP is just a way to choose which option policies to learn during training? The relationship/difference to previous work on hierarchical RL should be made clearer in the Related Work or Introduction section

**Questions:**

Questions/suggestions are included in the "Weaknesses" section. I am very open to raising my score if my concerns are addressed

---

> ### Author Response · Authors · 2025-11-21
>
> We sincerely appreciate your valuable feedback and suggestions.
>
> *Q1. Figure 2 and 3 are confusing. It’s not clear what the grey and black balls are supposed to represent.*
>
> A1. See lines 167-170: “In the top level, nodes represent abstract states and edges represent options. In the bottom level, nodes represent environment states and edges represent environment actions.”
>
> *Q2. Some analysis of sample/computational efficiency should be provided in the main text.*
>
> A2. Thanks for this great suggestion. In lines 295-303, we added a new table with results on sample and computational efficiency.
>
> *Q3. Some discussion of how this algorithm might be used when there is not access to a simulator should also be included.*
>
> A3. Access to a physics simulator is an explicit assumption of our setting (line 139): the user-provided abstract planner already requires a full simulator to define symbolic states and to execute its predefined skills, which is standard in Task and Motion Planning (TAMP) \[1\]. We agree that removing this assumption is an important direction for future work. We are planning to leverage recent Real2Sim2Real techniques \[2,3\] to reconstruct approximate simulators from real-world data and learn shortcut policies within those reconstructed environments.
>
> *Q4. I am unsure of the novelty of the approach.*
>
> A4. See the initial global response.
>
> Do these changes fully address your feedback about the paper? If not, we look forward to continuing the discussion!
>
> \[1\] Caelan Reed Garrett, Rohan Chitnis, Rachel Holladay, Beomjoon Kim, Tom Silver, Leslie Pack Kaelbling, and Tomás Lozano-Pérez. Integrated task and motion planning. *Annual review of control, robotics, and autonomous systems*, 4(1):265–293, 2021\.
> \[2\] V. Lim, H. Huang, L. Y. Chen, J. Wang, J. Ichnowski, D. Seita, M. Laskey, and K. Goldberg, “Real2sim2real: Self-supervised learning of physical single-step dynamic actions for planar robot casting,” in *2022 International Conference on Robotics and Automation (ICRA)*, pp. 8282–8289, IEEE, 2022\.
> \[3\] Shaoting Zhu, Linzhan Mou, Derun Li, Baijun Ye, Runhan Huang, and Hang Zhao. Vr-robo: A real-to-sim-to-real framework for visual robot navigation and locomotion. *arXiv preprint arXiv:2502.01536*, 2025\.

---

> > ### Comment · Reviewer_sKAL · 2025-11-24
> >
> > I have updated my score because my main concern about novelty has been addressed. I still think that figure 2 and 3 should stand on their own with their captions without having to read the text, and that you should mention the physics simulator somewhere as future work.

---

> > > ### Author Response · Authors · 2025-11-25
> > >
> > > Thank you for engaging with our work and these further suggestions! We have updated the caption for Figure 2 (lines 175-177) and Section 6 (lines 510-514) in the paper accordingly.

---

### Official Review · Reviewer_aDVn · 2025-10-31

**Soundness:** 2
**Presentation:** 2
**Contribution:** 1
**Rating:** 2
**Confidence:** 4

**Summary:**

This paper proposes SLAP (Shortcut Learning for Abstract Planning), a method that integrates model-free reinforcement learning (RL) with task and motion planning (TAMP) to automatically discover shortcut policies between abstract states. The main idea is to augment a symbolic planning graph with additional edges learned via RL. Each shortcut policy is trained to transition directly between two abstract states which effectively bypasses multiple intermediate actions and reduces plan length.

**Strengths:**

- The intuition of the paper (discovering shortcut connections in an abstract planning graph) is conceptually sound and easy to grasp. It is a simple yet effective idea that directly addresses the inefficiency of long hierarchical plans in TAMP frameworks.

- The approach can produce genuinely new high-level actions that are not part of the manually defined option set (e.g., slap). This demonstrates the potential of the method to extend the agent’s action set beyond what is explicitly encoded by human designers.

- The paper is clearly written and well-structured, making the overall framework and experiments easy to follow.

- Long horizon is one of the reasons why RL does not work effectively. Solving small RL problems with short horizons makes sense.

**Weaknesses:**

A primary weakness of this work lies in its conceptual alignment with the TAMP paradigm and the choice of experimental baselines.

1. Conceptual Dissonance with TAMP: The core philosophy of TAMP is to find plans that are valid with respect to a given symbolic model, ensuring that high-level action sequences are grounded and physically feasible according to predefined rules. The proposed method, SLAP, learns "shortcut" policies (e.g., "slapping" a tower) that achieve a goal state by operating outside of this symbolic structure. While this can yield more efficient plans, it fundamentally reframes the problem from one of constrained, symbolic planning to one of unconstrained trajectory optimization. This raises the question of whether SLAP is improving upon a TAMP solution or solving a different, less constrained problem altogether. The potential for emergent, "destructive," or unpredictable behaviors is at odds with the safety and predictability that motivates the use of TAMP in the first place.

2. Inadequate Baselines: The comparison to model-free RL baselines is not particularly insightful. It is well-established that vanilla RL algorithms struggle with the long-horizon, sparse-reward problems that TAMP is designed to solve. Since SLAP heavily leverages the strong structural priors from the TAMP framework (the abstract graph and initial options), outperforming these baselines is an expected outcome and does not sufficiently isolate the contribution of the shortcut-learning mechanism itself.

In summary, the paper positions itself as a hybrid TAMP-RL method, yet it diverges from the core principles of TAMP and compares against RL methods on a task setup where they are known to be inefficient. A clearer articulation of its contribution, perhaps as a method for discovering new symbolic operators rather than just bypassing them, and a comparison against more relevant baselines would significantly strengthen the paper.

Below are some further remarks:
- The methodology depends on finding two abstract states where there is at least a success rate of K_{rollout}/N_{rollout} transitioning from one to another state by taking random actions. I find this very restrictive and dependent on the abstract planning graph generation.

- The paper lacks a clear and fair baseline. It does not make sense to use pure RL algorithms as baselines for such complex, long-horizon tasks that fundamentally require high-level task planning. Even in simpler motion-planning problems, RL methods typically demand more training steps than the 500k used here to achieve meaningful performance. Since SLAP is built on a TAMP framework with strong structural priors, comparisons to model-free RL is not ideal. That’s why it cannot be claimed that the proposed method increases the performance. The paper already includes a comparison with the no-shortcut case. But this is more of an ablation study rather than a baseline. Finally, reporting success rates over only 10 trials per task is insufficient for statistical reliability; larger-scale evaluations or confidence intervals are needed to support the claimed improvements.

- The core idea of using RL to learn shortcut policies between abstract states is not a significant conceptual contribution. The method essentially applies standard model-free RL within a predefined TAMP structure to learn transitions that skip multiple existing options. While this integration is practically useful, it does not introduce new algorithmic insights or theoretical advances in either reinforcement learning or task and motion planning. The contribution is therefore more of an engineering combination of known components than a novel methodological development.

- The notation in Section 4.4 is not intuitive. add(â) and del(â) denote sets of atoms, whereas rel(â) denotes a set of objects involved in those atoms. The subsequent use of expressions such as add(â_train)[σ(rel(â_train))] ⊆ add(â_eval) is confusing (the square-bracket substitution notation is unconventional and not clearly defined). I can understand what is meant but a more rigorous and transparent mathematical formulation (or pseudocode) of the object substitution process would significantly improve clarity.

- The paper claims that learned shortcut policies can be reused on new objects through an object-substitution mapping σ, which aligns the add/del atom sets after substitution. However, the method for finding σ is not clearly described. When multiple objects are involved, determining consistent multi-object mappings becomes nontrivial, yet the paper provides no explanation of how σ is computed, whether it must be one-to-one, or how ambiguities are resolved. This lack of detail makes it difficult to evaluate the reliability and scalability of the object substitution mechanism proposed in Section 4.4.

- Figure 4 is hard to read. I would create two figures: one with graphs, one showing a learned shortcut (and improve this image).

- The analysis for Q5 (“Which RL design decisions are important for learning shortcuts?”) is limited in scope. The section “Shortcut Policy Learning Analysis” examines only one factor (whether shortcut policies are trained independently or as a shared universal policy). Other key RL design aspects (e.g., algorithm choice, reward shaping, exploration strategy) are not explored. As a result, the section provides a partial answer to Q5 and does not fully justify the general phrasing of the question.

- The claimed generalization ability of SLAP is limited. Section 4.2 describes “generalization over objects,” but the mechanism is a simple object-substitution procedure based on symbolic equivalence of add/del atoms. This allows policy reuse only when new tasks are structurally isomorphic to training ones. The method does not learn to generalize over varying object geometries, dynamics, or unseen relational structures (its transfer is purely symbolic). Consequently, the generalization claim overstates the scope of what the approach can handle.

- The abstract claims that SLAP “consistently outperforms planning and RL baselines”. As Table 1 demonstrates, SLAP matches the Pure Planning baseline in success rate (100%) and only improves efficiency by reducing plan length. The improvement therefore lies in shorter trajectories, not higher task success.

**Questions:**

In addition to the remarks made for the Weaknesses section, here are some additional questions:

- In the pruning phase, a shortcut is only retained for RL training if random rollouts reach the terminal abstract state s_{\text{term}} in at least K_{\text{rollout}} / N_{\text{rollout}} (which is \approx 5% with the mentioned parameters) of attempts. Given that this requires non-negligible random success (especially for a 3-D environment) between abstract states, how are such reachable abstract-state pairs obtained? Is there any mechanism during the construction of the abstract planning graphs or the definition of abstract states that biases the graph toward physically close or feasible state pairs, making this 5 % success rate attainable? Because, even for the simplest 3-D tasks, having more than a 5% success rate by taking random actions is highly unlikely.

- My understanding is that Figure 4 shows how, as training progresses, more shortcut policies become reliable and therefore less likely to be excluded from use during inference. In other words, the number of shortcuts that remain usable in the evaluation graph increases as RL training improves their success rates. Is this interpretation correct? The current wording (“increasing the number of training steps leads to more shortcuts being successfully learned and incorporated as graph edges”) is somewhat unclear because the number of learned shortcuts does not change with training.

- “To create an initial state distribution, we do not assume that we can sample directly from sinit; instead, we sample from the states encountered in the abstract planning graphs for the training tasks.” I suppose the graph creation process is deterministic. So, doesn’t this limit the diversity of the initial state distribution? How robust is SLAP to out-of-distribution physical configurations at test time, and could additional randomization during graph construction improve generalization?

---

> ### Author Response · Authors · 2025-11-14
>
> We sincerely appreciate your thorough review and comments. Since your other main concern is the adequacy of our baselines, we address that here first and are happy to run additional experiments if needed. **We note that this is an initial clarification rather than our full rebuttal; we are sharing it early in case you would like us to run additional baselines or analyses.**
>
> We agree that because SLAP “heavily leverages the strong structural priors from the TAMP framework,” it is expected to outperform pure RL. For this reason, we included a **hierarchical RL baseline** that has access to all predefined option policies and outputs both low-level actions and skill-selection probabilities. To further favor this baseline, whenever it selects a predefined skill, we execute that skill to completion before returning control to the agent. As noted in lines 336-340 of “Methods Evaluated,” this baseline is designed to leverage the *same high-level priors* as SLAP.
>
> Because our work is the first to study improving the execution time of abstract planners, we acknowledge that the choice of directly comparable baselines is limited. Nevertheless, we incorporated the strongest components from existing methods, e.g., **HER-style goal relabeling** for sparse rewards and the aforementioned hierarchical RL baseline with full access to predefined skills.
>
> We hope this clarifies our baseline choices. If not, we welcome further discussion, and if you have specific methods or papers in mind, we are happy to add comparisons and revise the paper accordingly.

---

> ### Author Response · Authors · 2025-11-21
>
> We sincerely appreciate your valuable feedback and suggestions.
>
> *Q1. SLAP is conceptually dissonant with TAMP.*
>
> A1. See the global response. SLAP is philosophically similar to other work on action abstraction learning and TAMP. We updated Section 6 to mention limitations related to safety and predictability.
>
> *Q2. The comparison to model-free RL baselines is inadequate, since it is expected that vanilla RL algorithms struggle with the long-horizon, sparse-reward problems.*
>
> A2. See earlier in the thread: in addition to pure RL baselines, we have a hierarchical RL baseline that leverages the same high-level priors as SLAP. We also posted a global clarification addressing related concerns, and we committed to adding a stronger comparison using the SOL algorithm from Meta \[1\]. In our revision, SOL will be adapted to use the same predefined options as SLAP and intrinsic rewards for shortcut transitions, providing an even stronger additional baseline. We will report the results as soon as possible.
>
> *Q3. Random-rollout pruning mechanism is very restrictive and dependent on the abstract planning graph generation.*
>
> A3. We are not certain if we understand your concern about random-rollout pruning being “restrictive.” If you are concerned about potentially low success rates in random rollouts, please see the results and discussion in Appendix B.1 (mentioned in Section 4.2 in the main paper). In addition, random-rollout pruning depends only on the abstract planning graph itself and not the way in which the graph is generated (see Section 4.1). Let us know if we understand your point here or if we can clarify further.
>
> *Q4. Reporting success rates over only 10 trials per task is insufficient for statistical reliability; larger-scale evaluations or confidence intervals are needed to support the claimed improvements.*
>
> A4. To clarify, our original submission reported success rates and plan lengths over *50 trials*. For each environment, we sample 10 held-out tasks (line 376\) and evaluate each method with 5 random seeds, yielding 50 evaluation episodes per environment (10 tasks x 5 seeds; see Table 1 caption and line 411). During this rebuttal period, we increased to 10 random seeds and updated main results over *100 trials* in Table 1\. These numbers are consistent with or better than prior work. For example, the “Soft Actor-Critic” paper \[2\] uses 5 random seeds and only 1 trial per seed. Furthermore, in the original submission, we showed standard deviations for plan lengths but not for success rates because all approaches converge to either consistently solving a task or consistently failing it. For example, pure RL and hierarchical RL baselines fail to solve long-horizon tasks in PyBullet environments completely, while SLAP and Pure Planning preserve completeness guarantees. For clarity, we added standard deviations for success rates in the updated paper version.
>
> *Q5. Clarity: a more rigorous and transparent mathematical formulation (or pseudocode) of the object substitution process.*
>
> A5. We have revised Section 4.4 and added Appendix A.2 to clarify our object substitution mechanism for shortcut generalization through pseudo-code (Algorithm 3).
>
> *Q6. In object substitution, how is mapping σ computed? Is it one-to-one? How are ambiguities resolved when multiple objects are involved?*
>
> A6. Relevant details are specified in Algorithm 3\. In a nutshell, our object substitution mechanism searches over all type-compatible objects seen at evaluation, and selects *the first* injective mapping that preserves every substituted add/del atom. It suffices to execute a shortcut policy on one mapping at evaluation to check if the shortcut edge can be successfully added to the abstract planning graph, given that our RL shortcuts involve motions that deal with multiple objects at once. However, we agree that SLAP could be benefited by considering more candidate mappings for generalization. We are happy to run additional experiments if the results would be helpful.
>
> *Q7. Figure 4 is hard to read.*
>
> A7. We have revised Figure 4 and updated the paper.
>
> *Q8. “Shortcut Policy Learning Analysis” examines only one factor (whether shortcut policies are trained independently or as a shared universal policy). Other key RL design aspects (e.g., algorithm choice, reward shaping, exploration strategy) are not explored.*
>
> A8. We have revised Section 5.1 and pointed to ablation results in Appendix B.2 on different algorithmic backbones — PPO, SAC, and SAC+HER. They provide representative RL baselines for examining how SLAP shortcut learning is affected by different reward shaping and exploration strategies.

---

> ### Author Response · Authors · 2025-11-21
>
> *Q9. The method does not learn to generalize over varying object geometries, dynamics, or unseen relational structures (its transfer is purely symbolic). The generalization claim overstates the scope of what the approach can handle.*
>
> A9. SLAP generalizes over varying object geometries and dynamics, but not over unseen relational structures. Our use of symbolic equivalence to guide object substitution is intentional: SLAP reuses a shortcut policy whenever its expected symbolic add/del effects can be matched under a substitution. This design enables broad generalization — a single learned shortcut can transfer to *many different objects* and *many different numbers of objects* as long as their symbolic effects align. As for generalization over goals, it is a consequence of both object substitution and the structure of abstract planning graphs (see Appendix D.1). This supports our claim that “SLAP generalizes over tasks (initial states and goals) and objects” (line 235).
>
> To further support this generalization claim, we updated “Generalization Capability Analysis” and Figure 5 in Section 5.1 with more results on different object dynamics (mass and friction).
>
> *Q10. The abstract claims that SLAP “consistently outperforms planning and RL baselines”. As Table 1 demonstrates, SLAP only improves efficiency by reducing plan length. The improvement therefore lies in shorter trajectories, not higher task success.*
>
> A10. We stated our claims clearly in the abstract (line 19-21): “Without any additional assumptions or inputs, shortcut learning leads to shorter solutions than pure planning, and higher task success rates than flat and hierarchical RL.”
>
> *Q11. Random-rollout mechanism requires non-negligible random success, how are such reachable abstract-state pairs obtained?*
>
> A11. As described in Section 4.1 and Appendix A.1, expanding the top level of the abstract planning graph requires only checking which predefined options are feasible from the current abstract state. We provided empirical evidence on the effectiveness of the random-rollout mechanism in finding promising shortcuts in Appendix B.1 (mentioned in line 215, Section 4.2).
>
> *Q12. Is this interpretation correct: the number of shortcuts that remain usable in the evaluation graph increases as RL training improves their success rates?*
>
> A12. Yes. We have revised the wording in “Training Steps Analysis” of Section 5.1.
>
> *Q13. Line 204-208: “To create an initial state distribution, we do not assume that we can sample directly from sinit; instead, we sample from the states encountered in the abstract planning graphs for the training tasks.” Doesn’t this limit the diversity of the initial state distribution? How robust is SLAP to out-of-distribution physical configurations at test time, and could additional randomization during graph construction improve generalization?*
>
> A13. We agree that this may limit diversity, but *assuming access to the true initial-state distribution for every shortcut MDP would be unrealistically strong*. And in practice, the low-level states collected from abstract planning graphs are reasonably diverse, since multiple option sequences (paths in the graphs) often reach the same abstract states.
>
> We added new ablations in Appendix D.5 comparing SLAP’s performance when evaluated directly on out-of-distribution physical configurations versus when those configurations are incorporated into the shortcut training process.
>
> Do these changes fully address your feedback about the paper? If not, we look forward to continuing the discussion!
>
> \[1\] Mikael Henaff, Scott Fujimoto, Michael Matthews, and Michael Rabbat. Scalable Option Learning in High-Throughput Environments. *arXiv preprint arXiv:2509.00338*, 2025\.
> \[2\] Tuomas Haarnoja, Aurick Zhou, Pieter Abbeel, and Sergey Levine. Soft actor-critic: Off-policy maximum entropy deep reinforcement learning with a stochastic actor. In *International conference on machine learning*, pp. 1861–1870. Pmlr, 2018\.

---

### Official Review · Reviewer_bFSp · 2025-11-01

**Soundness:** 3
**Presentation:** 3
**Contribution:** 2
**Rating:** 6
**Confidence:** 4

**Summary:**

This paper proposes an approach to facilitate Task and Motion Planning efficiency (in term of plan length) by learning additional options to get across abstract states using RL. Compared to simply using a set of limited options (e.g., move the block tower one by one), the proposed method enriches the set of options and opens the possibility of reaching goal states with few steps (e.g., push down the tower entirely).

**Strengths:**

The paper is well written and easy to read, especially Fig 1 provides a good illustration of desired shortcuts and Fig 3 gives a clear overview of the method.

Technically, the proposed method to generalize over objects / object numbers, to the best of my knowledge, provides an interesting and novel way for downstream adaptation.

Additionally, the experiment results strongly support the proposed method, where it outperforms vanilla TAMP and hierachical RL methods.

**Weaknesses:**

My major question is around the novelty of the proposed method. There are many skill learning methods where people learn skills by specifying a goal state (and optionally a start state) with RL, and then apply such learned skills to downstream planning or HRL. If I understand correctly, the proposed method is very similar to those methods, except that the start and goal states come from the planning graph. Is that right? Any novelty I missed?

Meanwhile, I wonder what if the learned shortcut has non-zero failure rates, will SLAP still adopts such options? If so, what if the RL learned options fail will lead to an unrecoverable state, will the planning take such possibility into account?

**Questions:**

When conducting random rollout pruning, what is the action space, atom action or provided options?

---

> ### Author Response · Authors · 2025-11-21
>
> We sincerely appreciate your valuable feedback and suggestions.
>
> *Q1. Novelty of the proposed method?*
>
> A1. See the initial global response.
>
> *Q2. If the learned shortcut has non-zero failure rates, will SLAP still adopt such options? If so, what if the RL learned options fail will lead to an unrecoverable state, will the planning take such possibility into account?*
>
> A2. At evaluation time, SLAP only uses shortcuts whose terminal abstract states can be reliably reached. As mentioned in line 241 of Section 4.3, we check if the abstract terminal state of a shortcut can be reached within $T\_{\\text{eval}}$ steps and prune the edges in the case of failure. Importantly, planning is performed over the abstract graph, so if a shortcut leads to an unexpected or unrecoverable continuous state, the planner can simply backtrack and choose an alternative path.
>
> *Q3. When conducting random rollout pruning, what is the action space, atom action or provided options?*
>
> A3. Random rollout pruning takes long sequences of actions in the low-level (atomic) action space (lines 211-215, the same action space used by the shortcut policies). Random rollout pruning doesn’t use the provided options.
>
> Do these changes fully address your feedback about the paper? If not, we look forward to continuing the discussion!

---

> > ### Comment · Reviewer_bFSp · 2025-11-26
> >
> > I appreciate the detailed explanation from the authors. All my concerns are resolved. I view the paper positively as before and will keep my score the same.

---

### Official Review · Reviewer_AFvJ · 2025-11-01

**Soundness:** 3
**Presentation:** 3
**Contribution:** 3
**Rating:** 8
**Confidence:** 3

**Summary:**

This paper tackles long horizon decision making in robotics where classic task and motion planning relies on hand specified skills and often yields long satisficing plans. The authors propose Shortcut Learning for Abstract Planning, or SLAP, which automatically learns new low level options that act as shortcuts between abstract states in a planning graph induced by the existing skill library. The system builds a two level abstract planning graph, searches for shortest executions at the low level, then augments the graph with learned options trained through model free RL in self contained shortcut MDPs with step penalties and goal termination. A simple but effective pruning strategy screens shortcut candidates using random rollouts before launching PPO training. At test time, SLAP plans with both given skills and learned options, prunes failing edges online, and selects shorter plans via Dijkstra on the ground graph. To generalize beyond the training object set, SLAP projects observations onto relevant objects determined from add and delete atom sets and applies object substitution to reuse learned policies when object identities differ. This preserves the relational inductive bias of TAMP while enabling dynamic behaviors beyond fingertip grasp and place such as slap, wiggle, and wipe.

**Strengths:**

Framing and practical value
- Clear problem statement that existing TAMP systems rely on hand designed skills which limits efficiency and expressivity. SLAP focuses squarely on reducing execution time without discarding the benefits of abstraction and search.
- Elegant algorithmic design that keeps planning and learning modular. The abstract planning graph yields a well defined search problem, and shortcuts are learned in parallel MDPs with simple goal conditions and dense step penalties.
- Relational generalization via relevant object projection and object substitution is well motivated and effective, letting the same shortcut policy apply across object sets and counts.

Empirical evidence
- Consistent improvements in plan length across four diverse domains with long horizons and sparse rewards. Reductions are large in magnitude, for example about 69 percent in Obstacle Tower and over 50 percent in the PyBullet tasks.
- Clear wins over strong baselines representing three regimes: pure planning, pure RL with PPO and SAC plus HER, and hierarchical RL with access to the same predefined skills.
- Training steps analysis and shortcut discovery counts show monotonic gains as more shortcuts are learned, connecting learning progress to planning improvements.
- Generalization experiments demonstrate robustness to different numbers of obstacles and to additional distractor objects. The learned dynamic behaviors like slap and wipe act on multiple objects and keep plan lengths stable.

**Weaknesses:**

Assumptions and scope
- Relies on a known transition function and fully observable deterministic settings for graph construction, although variants relax these assumptions. Real systems often face sensing delays, latency, and controller noise which may require tighter integration of failure recovery and uncertainty aware planning.
- Assumes the provided option set enables task completion. When this is not true, completeness can be lost. Appendix results discuss such cases but a stronger treatment of failure detection and fallback would help.

Scalability and compute
- The number of candidate shortcuts scales with the square of the number of abstract states. The pruning heuristic is simple and effective but a learned prioritizer or search over the shortcut space could further reduce training cost. Wall clock budgets and GPU hours per environment would make the compute footprint transparent.
- Planning time modestly increases compared to pure planning in two domains before being outweighed by shorter execution. A more thorough analysis of search heuristics and graph size versus latency would help practitioners choose configurations.

**Questions:**

- How many shortcut policies were ultimately trained and kept per environment during the runs reported in Table 1, and what was the total wall clock training time per environment including pruning and PPO updates?
- What termination condition is used for each learned option at test time beyond the abstract state check and step limit. Do you implement safety guards such as maximum end effector velocity or minimum clearance during dynamic actions like a slap or wipe?
- You evaluate separate PPO policies per shortcut pair. How do results compare to a single goal conditioned policy trained across shortcut goals when both are tuned equally, and how does data efficiency change as the number of abstract states grows?
- The object substitution criterion checks add and delete inclusions. How is the mapping computed in practice when multiple candidates exist, and what is its computational cost during planning on the larger PyBullet graphs?
- For hierarchical RL, did you try curriculum learning, intrinsic motivation, or hindsight relabeling at the high level to mitigate sparse rewards before concluding failure on the three PyBullet domains?

---

> ### Author Response · Authors · 2025-11-21
>
> We sincerely appreciate your valuable feedback and suggestions.
>
> *Q1. Real systems often face sensing delays, latency, and controller noise which may require tighter integration of failure recovery and uncertainty-aware planning.*
>
> A1. Thanks for the suggestion\! In ongoing follow-up work, we are considering exactly these challenges, looking at how we can combine SLAP with uncertainty-aware planning and failure recovery to improve performance in real-world robotic tasks.
>
> *Q2. Wall clock budgets and GPU hours per environment would make the compute footprint transparent.*
>
> A2. Thanks for this great suggestion\! In lines 295-303, we added a new table with results on sample and computational efficiency.
>
> *Q3. A more thorough analysis of search heuristics and graph size versus latency would help practitioners choose configurations.*
>
> A3. In SLAP, planning latency is dominated not by the *size* of the abstract graph, but by how quickly Dijkstra can identify a low-cost path containing a highly effective shortcut. Even in environments with large planning graphs (e.g., Obstacle Tower with 159 nodes and 324 edges), the presence of a strong shortcut dramatically shortens the optimal path (\~70 low-level steps). As shown in Appendix D.2, this causes Dijkstra to terminate early at test time, making SLAP’s planning time even shorter than Pure Planning.
>
> For practitioners, the most important configuration choice is therefore the reliability and quality of shortcut policies, not the graph size or search heuristic. In domains where planning graphs are large but the agent learns strong shortcuts, SLAP’s planning latency remains low.
>
> *Q4. How many shortcut policies were ultimately trained and kept per environment during the runs reported in Table 1, and what was the total wall clock training time per environment including pruning and PPO updates?*
>
> A4. Through offline data collection and pruning before any RL training, we obtained a collection of 11 shortcuts in Obstacle 2D, 92 in Obstacle Tower, 74 in Cluttered Drawer, and 54 in Cleanup Table (lines 362-363). Towards the end of 500,000 training steps, the number of shortcuts successfully incorporated as graph edges are $4.4 \pm 0.8$ in Obstacle 2D, $17.6 \pm 2.1$ in Obstacle Tower, $17.0 \pm 1.2$ in Cluttered Drawer, and $13.4 \pm 1.5$ in Cleanup Table — i.e. last data points in training dynamics curves in Figure 4\. The total wall clock training time is reported in lines 295-303 in the compute table.
>
> *Q5. For termination conditions, do you implement safety guards such as maximum end effector velocity or minimum clearance?*
>
> A5. Yes, we impose limits on joint velocity and we opt for delta position control rather than velocity or torque control. We do not enforce minimum clearance because we want to allow the robot to make contact with objects (e.g., pushing with its arm). Like other work that uses RL with real robots, we would take additional measures to ensure safety before running on a real robot.
>
> *Q6. How do results compare to a single goal-conditioned policy trained across shortcut goals?*
>
> A6. We ablated shortcut policy learning schemes in Section 5.2 under “Shortcut Policy Learning Analysis”. We developed methods such as Abstract Subgoals and Abstract HER that are universal goal-conditioned shortcut policies. Compared to the independent shortcut policies we adopt in SLAP, they have comparably worse performance.
>
> *Q7. How is the mapping of object substitution computed when multiple candidates exist, and computational cost?*
>
> A7. We have revised Section 4.4 and added Appendix A.2 to clarify our object substitution for shortcut generalization mechanism through pseudo-code (Algorithm 3).
>
> In a nutshell, our object substitution mechanism searches over all type-compatible objects seen at evaluation, and selects *the first* injective mapping that preserves every substituted add/del atom. It suffices to execute a shortcut policy on one mapping at evaluation to check if the shortcut edge can be successfully added to the abstract planning graph, given that our RL shortcuts involve motions that deal with multiple objects at once. However, we agree that SLAP could be benefited by considering more candidate mappings for generalization.

---

> ### Author Response · Authors · 2025-11-21
>
> *Q8. For hierarchical RL, did you try curriculum learning, intrinsic motivation, or hindsight relabeling?*
>
> A8. Yes, we tried hierarchical SAC+HER, but same as PPO it achieved 0% success in all our PyBullet environments. Since HER did not improve option learning or skill selection on predefined options, we report only the PPO-based hierarchical RL baseline.
>
> We recently posted a global clarification addressing concerns about hierarchical RL baselines, where we committed to adding a stronger comparison using the SOL algorithm from Meta \[1\]. In our revision, SOL will be adapted to use the same predefined options as SLAP and intrinsic rewards for shortcut transitions, providing an even stronger additional baseline. We will report the results as soon as possible.
>
> Do these changes fully address your feedback about the paper? If not, we look forward to continuing the discussion!
>
> \[1\] Mikael Henaff, Scott Fujimoto, Michael Matthews, and Michael Rabbat. Scalable Option Learning in High-Throughput Environments. *arXiv preprint arXiv:2509.00338*, 2025\.

---

### Author Response · Authors · 2025-11-14

Dear Reviewers and Area Chair,

Thank you for your thoughtful feedback. We provide an initial clarification here and welcome any further questions or requests for analyses. **This is not our full rebuttal; it is an early post intended to confirm what additional experiments or clarifications would be most helpful.**

First, regarding novelty (**sKAL** and **bFSp**): our work is **the first** to study the problem setting of **improving the execution time of an abstract planner**. This is an important problem because TAMP plans are often long and inefficient due to simplifying assumptions about physical interactions. Our contribution is not a new TAMP or RL algorithm, but a framework that **combines planning and RL to autonomously learn shortcuts that reduce execution time**. This is stated in lines 449-50, 82-83, 92-93 of Introduction, and further distinguished from related learning-for-TAMP work in lines 108-121.

**On sKAL’s question**: “It seems like SLAP is just a way to choose which option policies to learn during training?”

Much of the literature on option discovery is fundamentally addressing the question of which options to learn during training \[1-6\]. SLAP proposes a novel answer to this question. It also (1) improves the execution time of an abstract planner; (2) integrates learned and predefined skills to produce shorter plans; and (3) generalizes to new goals and different numbers of objects using planning.

We kindly ask reviewers sKAL and bFSp: do these clarifications resolve your concerns about novelty? If not, we are happy to discuss further and to add comparisons to any specific methods you have in mind.

**On reviewer aDVn’s concern about SLAP’s “conceptual dissonance” with TAMP**:

Respectfully, we have a different perspective on TAMP and acknowledge that there is not necessarily a consensus in the TAMP community on this point. Like other papers that combine action abstraction learning and TAMP \[e.g., 7-10\] (cited in lines 111-113), our ultimate goal is to solve long-horizon decision-making problems efficiently and effectively. SLAP’s ability to discover shortcuts that are “outside the symbolic structure”—beyond what we know how to manually engineer—is a key strength from this perspective. However, we agree that the behavior of SLAP (as well as other combinations of learning and TAMP) is less predictable than TAMP alone, and we will update the paper to acknowledge this limitation.

\[1\] Tejas D Kulkarni, Karthik Narasimhan, Ardavan Saeedi, and Josh Tenenbaum. Hierarchical deep reinforcement learning: Integrating temporal abstraction and intrinsic motivation. *Advances in neural information processing systems*, 29, 2016\.
\[2\] Marcin Andrychowicz, Filip Wolski, Alex Ray, Jonas Schneider, Rachel Fong, Peter Welinder, Bob McGrew, Josh Tobin, OpenAI Pieter Abbeel, and Wojciech Zaremba. Hindsight experience replay. *Advances in neural information processing systems*, 30, 2017\.
\[3\] Ofir Nachum, Shixiang Shane Gu, Honglak Lee, and Sergey Levine. Data-efficient hierarchical reinforcement learning. *Advances in neural information processing systems*, 31, 2018\.
\[4\] Tuomas Haarnoja, Aurick Zhou, Pieter Abbeel, and Sergey Levine. Soft actor-critic: Off-policy maximum entropy deep reinforcement learning with a stochastic actor. In *International conference on machine learning*, pp. 1861–1870. Pmlr, 2018\.
\[5\] Ben Eysenbach, Russ R Salakhutdinov, and Sergey Levine. Search on the replay buffer: Bridging planning and reinforcement learning. *Advances in neural information processing systems*, 32, 2019\.
\[6\] Nikolay Savinov, Alexey Dosovitskiy, and Vladlen Koltun. Semi-parametric topological memory for navigation. *arXiv preprint arXiv:1803.00653*, 2018\.
\[7\] Tom Silver, Rohan Chitnis, Joshua Tenenbaum, Leslie Pack Kaelbling, and Tomás Lozano-Pérez. Learning Symbolic Operators for Task and Motion Planning. In *Proceedings of the IEEE/RSJ International Conference on Intelligent Robots and Systems (IROS)*, pp. 3182–3189, 2021b.
\[8\] Shuo Cheng and Danfei Xu. League: Guided skill learning and abstraction for long-horizon manipulation. *IEEE Robotics and Automation Letters*, 8(10):6451–6458, 2023\.
\[9\] Christopher Agia, Toki Migimatsu, Jiajun Wu, and Jeannette Bohg. Stap: Sequencing task-agnostic policies. In *2023 IEEE International Conference on Robotics and Automation (ICRA)*, pp. 7951– 7958\. IEEE, 2023\.
\[10\] Ajay Mandlekar, Caelan Reed Garrett, Danfei Xu, and Dieter Fox. Human-in-the-loop task and motion planning for imitation learning. In *Conference on Robot Learning*, pp. 3030–3060. PMLR, 2023\.

---

> ### Comment · Reviewer_sKAL · 2025-11-15
> **Response to rebuttal about novelty**
>
> I first would like to say that I am unfamiliar with all the relevant literature so may be mistaken in my assessment.
>
> I think my main concern is about comparisons to hierarchical RL/option discovery methods. Is it fair to say that your method is similar to option discovery methods but using the user pre-defined options as the base actions? If so, can you compare to this baseline?

---

> > ### Author Response · Authors · 2025-11-15
> >
> > Thank you for your follow-up and for engaging with our work. We would like to clarify the relationship between SLAP and hierarchical RL/option-discovery methods.
> >
> > **Comparison to hierarchical RL.**
> > As reported in Section 5 (“Methods Evaluated”), we have compared against a hierarchical RL baseline that **has access to exactly the same user-provided options as SLAP**, and SLAP consistently achieves shorter plans. This baseline outputs both low-level actions and skill-selection probabilities; to favor this baseline, whenever it chooses a predefined option, we execute that option to completion.
> >
> > To further strengthen our comparisons, we will also include a **new hierarchical RL baseline** using **SOL** from Meta \[1\]. While SOL was not designed to discover shortcuts in the sense studied by SLAP, it can learn new option policies via intrinsic rewards tied to particular transitions. We will (i) provide SOL with the same predefined option policies as SLAP and our existing hierarchical RL baseline, and (ii) define intrinsic rewards for each shortcut transition so that SOL can learn corresponding low-level options. This produces a strong hierarchical RL baseline: SOL’s high-level policy selects among both predefined and newly learned options, but—unlike SLAP—it does not perform planning over abstract states and does not support object substitution or abstract-state generalization. We believe this is a fair and informative comparison, and **we will include this SOL baseline in our revised submission**.
> >
> > **Relation to option-discovery methods.**
> > Algorithmically, SLAP is indeed inspired by ideas from option discovery—e.g., identifying useful transitions/subgoals and learning policies between them. However, **SLAP is designed to tackle a different problem.** Most option-discovery approaches \[2-4\] aim to make long-horizon tasks solvable by discovering subgoals or abstractions from interaction data. In contrast, **SLAP assumes that the given options already make planning feasible**, and focuses instead on a **new objective: improving the *execution time* of existing plans through low-level shortcuts.** To our knowledge, no prior option-discovery method is designed for or evaluated in this setting. We view more advanced option-discovery techniques as complementary and as a promising direction for future work.
> >
> > We hope this clarifies the distinction, and we are happy to run additional comparisons if the reviewer has specific baselines in mind.
> >
> > \[1\] Mikael Henaff, Scott Fujimoto, Michael Matthews, and Michael Rabbat. Scalable Option Learning in High-Throughput Environments. *arXiv preprint arXiv:2509.00338*, 2025\.
> > \[2\] Cameron Allen, Timo Gros, Michael Katz, Harsha Kokel, Hector Palacios, and Sarath Sreedharan. Bridging the gap between ai planning and reinforcement learning (prl@ ijcai 2023). In *International Joint Conference on Artificial Intelligence*, 2023\.
> > \[3\] Ben Eysenbach, Russ R Salakhutdinov, and Sergey Levine. Search on the replay buffer: Bridging planning and reinforcement learning. *Advances in neural information processing systems*, 32, 2019\.
> > \[4\] Nikolay Savinov, Alexey Dosovitskiy, and Vladlen Koltun. Semi-parametric topological memory for navigation. *arXiv preprint arXiv:1803.00653*, 2018\.

---

> > > ### Comment · Reviewer_sKAL · 2025-11-24
> > >
> > > I thank the authors for the clarification and have updated my score accordingly

---

### Author Response · Authors · 2025-12-03

Dear Area Chair and Reviewers,

As promised in the previous global response, we have implemented a new hierarchical RL baseline with SOL \[1\]. Throughout the discussion period, we ablated different design choices, made modifications to the original implementation, presented additional inputs to the algorithm to make our long-horizon tasks easier for it to learn, and performed comprehensive hyperparameter tuning.

The results conducted over 10 seeds have been reported in Table 1 in the updated paper version, with explanations of our SOL adaptation and hyperparameters in lines 347-352, 371-375 of Section 5 and Appendix C.

Note: Due to modified content in this updated paper version, the line numbers referenced in our discussion threads with reviewers may no longer be accurate. Please refer to the previous paper version accordingly. Thank you\!

\[1\] Mikael Henaff, Scott Fujimoto, Michael Matthews, and Michael Rabbat. Scalable Option Learning in High-Throughput Environments. *arXiv preprint arXiv:2509.00338*, 2025\.

---

### Author Response · Authors · 2025-12-03
**Summary**

Dear Area Chair and Reviewers,

We sincerely thank you for your valuable time and efforts. We are pleased to see that **most reviewers have responded positively** to our work. Below, we summarize the main merits and concerns raised by the reviewers, along with our responses to these concerns during the discussion period. For detailed experimental results, please see the corresponding Reviewer Response thread.

Novelty and Significance
The main contribution and novelty of our work are:

* We introduce SLAP, an approach that combines Task and Motion Planning (TAMP) and Reinforcement Learning (RL) to tackle **the problem of inefficient execution times of TAMP methods** which often have limiting predefined skills that make strong simplifying assumptions about robot-object interactions. To the best of our knowledge, our work is **the first** to use RL to improve the execution time of an abstract planner, **by up to 73%**.
* SLAP **automatically defines shortcut MDPs for RL** to learn by leveraging the **abstract planning graph** induced by the predefined skills. This addresses RL’s limited performance in long-horizon continuous robotics tasks with sparse rewards.

As noted by Reviewer AFvJ, SLAP has “an elegant algorithmic design that keeps planning and learning modular — the abstract planning graph yields a well defined search problem, and shortcuts are learned in parallel MDPs with simple goal conditions and dense step penalties.” Some reviewers originally had questions on SLAP’s novelty, specifically its connection to literature on option discovery. We emphasized in the early global response that SLAP is the first to tackle the problem setting of improving the execution time of an abstract planner. Accordingly, we are encouraged to hear from reviewers indicating that this concern has been addressed: Reviewer sKAL **raised their score from 4 to 6**, and Reviewer bFSp mentioned they **view the paper positively as before and keep score 6**.

Reviewers AFvJ and bFSp also noted that SLAP’s “relational generalization via relevant object projection and object substitution is well motivated and effective.” To address related clarity concerns of Reviewer aDVn and AFvJ, we updated the paper with mathematical formulation in Section 4.4 and Algorithm 3 pseudo-code in Appendix A.2.

Experiments
We are encouraged to see that most reviewers agree “the experiment results strongly support the proposed method, where it outperforms vanilla TAMP and hierarchical RL methods” (Reviewer bFSp, sKAL); SLAP achieves “consistent and large improvements in plan length across four diverse domains with long horizons and sparse rewards” (Reviewer AFvJ), “enriches the set of options and opens the possibility of reaching goal states with few steps (e.g., push down the tower entirely)” (Reviewer bFSp).

To best address some of the reviewers’ concerns on our comparisons to hierarchical RL/option discovery methods (Reviewer sKAL, global response; Reviewer AFvJ, Q\&A 8), we implemented **a new, stronger hierarchical RL baseline** using **SOL** from Meta throughout the discussion period. See the last global response we posted.

To make our computational footprint more transparent as mentioned by Reviewers sKAL and AFvJ, we added a new table with results on sample and computational efficiency in lines 295-303. We also modified Figure 4 and the caption of Figure 2 for better clarity according to Reviewer aDVn and sKAL’s advice.

Throughout the rebuttals, we conducted extensive ablations to answer Reviewer aDVn’s questions on our algorithm design choices and SLAP’s robustness. We experimented with different shortcut policy backbones (PPO, SAC, and SAC+HER) to ablate on how shortcut learning is affected by different reward shaping and exploration strategies (Reviewer aDVn, Q\&A 8, reported in Appendix B.2). To assess robustness, we evaluated SLAP on out-of-distribution physical configurations at test time (Reviewer aDVn, Q\&A 13, reported in Appendix D.5), as well as on varying number of objects with different geometries during SLAP’s relational generalization with object substitution (Reviewer aDVn, Q\&A 9, reported in Figure 5 of Section 5.1).

Thank you all again for helping us improve and strengthen our work.

---

### Meta-Review · Area_Chair_34Sb · 2026-01-06

**Summary:**

This paper proposes SLAP (Shortcut Learning for Abstract Planning), a method that integrates model-free reinforcement learning (RL) with task and motion planning (TAMP) to automatically discover shortcut policies between abstract states. The main idea is to augment a symbolic planning graph with additional edges learned via RL. Each shortcut policy is trained to transition directly between two abstract states, which effectively bypasses multiple intermediate actions and reduces plan length.

The reviewers' main concerns focus on Inadequate Baselines and Conceptual Dissonance with TAMP, as well as missing analysis of sample/computational efficiency.

The authors provide a detailed response to all reviewers, which may well address the reviewers' concerns.

The scores are 8, 6, 4, 2. After the rebuttal,  Reviewer sKAL updated the score from 4 to 6 (Borderline Accept).

**Reviewer Concerns:**

The reviewers' main concerns focus on Inadequate Baselines and Conceptual Dissonance with TAMP, as well as missing analysis of sample/computational efficiency.

The authors provide a detailed response to reviewers with a clarification about the Comparison to hierarchical RL and the Relation to option-discovery methods, which may address the Reviewer aDVn 's and Reviewer sKAL's concern.

**Reviewer Scores:**

The scores are 8, 6, 4, 2. After the rebuttal,  Reviewer sKAL updated the score from 4 to 6 (Borderline Accept).

---

### Decision · Program_Chairs · 2026-01-26

Accept (Poster)